# Pan-Antarctic analysis aggregating spatial estimates of Adélie penguin abundance reveals robust dynamics despite stochastic noise

Christian Che-Castaldo [1], Stephanie Jenouvrier [2,3], Casey Youngflesh [1], Kevin T. Shoemaker [1,4], Grant Humphries [1,5], Philip McDowall[1], Laura Landrum [6], Marika M. Holland[6], Yun Li [7,8], Rubao Ji[8] & Heather J. Lynch [1]

Colonially-breeding seabirds have long served as indicator species for the health of the oceans on which they depend. Abundance and breeding data are repeatedly collected at fixed study sites in the hopes that changes in abundance and productivity may be useful for adaptive management of marine resources, but their suitability for this purpose is often unknown. To address this, we fit a Bayesian population dynamics model that includes process and observation error to all known Adélie penguin abundance data (1982–2015) in the Antarctic, covering >95% of their population globally. We find that process error exceeds observation error in this system, and that continent-wide "year effects" strongly influence population growth rates. Our findings have important implications for the use of Adélie penguins in Southern Ocean feedback management, and suggest that aggregating abundance across space provides the fastest reliable signal of true population change for species whose dynamics are driven by stochastic processes.

[1] Department of Ecology & Evolution, Stony Brook University, Life Sciences 106, Stony Brook, NY 11794, USA. [2] Biology Department, Mailstop 50, Woods Hole Oceanographic Institution, 266 Woods Hole Road, Woods Hole, MA 02543, USA. [3] Centre d'Etudes Biologiques de Chize, UMR 7372 CNRS/University La Rochelle, 79360 Villiers en Bois, France. [4] Department of Natural Resources and Environmental Science, University of Nevada, 1664 N. Virginia Street, Reno, NV 89557, USA. [5] Black Bawks Data Science Ltd, 24 Abertarff Place, Fort Augustus PH32 4DR, UK. [6] National Center for Atmospheric Research, P.O. Box 3000, Boulder, CO 80307, USA. [7] College of Marine Science, University of South Florida, 140 7th Avenue South, St. Petersburg, FL 33701, USA. [8] Biology Department, Mailstop 33, Redfield 2-14, Woods Hole Oceanographic Institution, 266 Woods Hole Road, Woods Hole, MA 02543, USA. Correspondence and requests for materials should be addressed to C.C.-C. (email: christian.che-castaldo@stonybrook.edu) or to H.J.L. (email: heather.lynch@stonybrook.edu)

Seabirds are widely considered reliable environmental indicators, as they are sensitive to multiple terrestrial and marine environmental factors[1-3]. In the Southern Ocean, Adélie penguins have played a critical role in understanding how the Antarctic ecosystem is changing in response to climate change, and this information has been key in shaping Antarctic conservation policy and management decisions[4-7]. However, despite the community's reliance on the Adélie penguin as a marine sentinel[8], there has been a nearly 30-year debate over the key mechanisms underlying Adélie population dynamics. The root of this uncertainty involves the relative influence of, and interactions among, factors known to influence Adélie abundance, such as prey and nest availability, sea ice conditions, predation, and weather[9-16]. There have been many recent efforts to understand what drives observed changes in Adélie abundance and distribution (e.g., refs. [17-20]) and several recent studies have identified clear trends in abundance within specific regions[12, 18-23].

However, despite significant advances in understanding long-term trends, much remains unknown about what drives inter-annual variability and whether such fluctuations are informative of underlying ecosystem dynamics. Given the current limits of data and covariate availability, the magnitude of process error (i.e., unexplained variation in true abundance driven by unmodeled biotic or abiotic processes) and its dominance over observation error (i.e., errors due to measurement imprecision) is gaining recognition as an important driver of population dynamics, both in the Antarctic[24, 25] and in other systems[26]. Moreover, the relative paucity of complete time series has made it difficult to generate robust metrics of environmental change through spatial or temporal aggregation, an idea initially raised by the Commission for the Conservation of Antarctic Marine Living Resources[24] (CCAMLR), which was established with the objective of conserving and managing Antarctic marine life. While CCAMLR has recognized that both stochastic and deterministic influences must be addressed when designing an adaptive feedback management system, tools for doing so remain an area of active development in statistical ecology and have not been fully utilized in the Antarctic. Addressing both the temporal and spatial components of Adélie population change will not only facilitate the use of the Adélie penguin as a tool for feedback management, but will allow for better model predictions on the abundance and distribution of Adélie penguins under climate change[27].

Using all the publicly available data on Adélie penguin abundance and distribution since 1982, we model population growth for each of Antarctica's 267 known Adélie penguin colonies using a hierarchical Bayesian model that includes both a deterministic component driven by environmental covariates and year-specific stochastic variation (see Methods section and Supplementary Data 1–3). This model allows us to partition variation in Adélie population growth rates and determine the magnitude and importance of process error in driving Adélie population fluctuations. We use these parameter estimates to simulate time series that demonstrate the extent to which short-term fluctuations in abundance may be considered unreliable measures of ecosystem health and quantify the spatial and temporal aggregation required to reliably detect Adélie population trends.

## Results

**Population growth rates and aggregated dynamics.** Population growth rate was positively associated with peak winter sea ice over the previous 4 years or, put another way, negatively associated with prolonged periods without extensive sea ice ("ice droughts") (Fig. 1a, c–f). Growth rate was also weakly correlated with extensive summer sea ice 4 years prior, with a posterior distribution that was negative but overlapped zero (Fig. 1a, c–f).

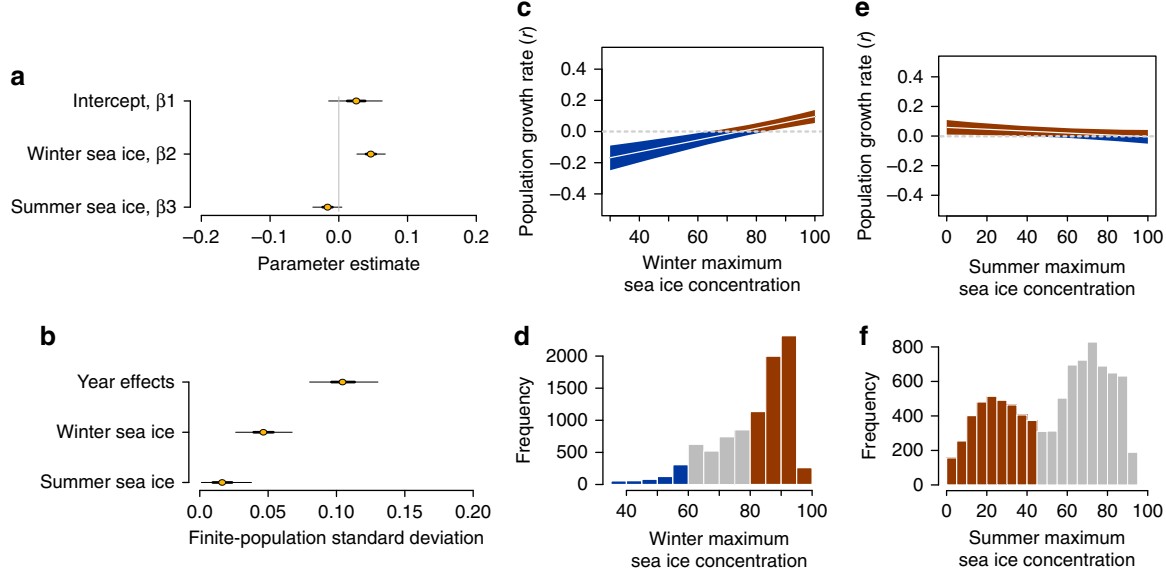

**Fig. 1** Parameter estimates and growth rate as a function of model covariates. **a** Adélie population growth rate (*r*) at average winter and summer peak sea ice concentrations ($\beta_1$), and the effect of a one standard deviation change in winter ($\beta_2$) or summer ($\beta_3$) peak sea ice concentration on the population growth rate. **b** The finite-population standard deviation for each source of variation in the Adélie intrinsic growth rate. For **a**, **b**, *thick lines* represent the 50% equal-tailed credible intervals, *thin lines* represent the 95% equal-tailed credible intervals, and *circles* are the posterior medians. **c**, **e** Adélie population growth rate as a function of winter (summer) peak sea ice concentration, at mean winter (summer) peak sea ice concentration. The *gray shaded areas* represent the 95% equal-tailed credible intervals and the *lines* are the posterior medians. **d**, **f** Histograms of actual winter and summer peak sea ice conditions for all 267 sites across 34 years. *Bars* have been color coded to represent values associated with positive (*red*) or negative (*blue*) growth rates; *gray bars* reflect values associated with credible intervals that include support for both positive and negative growth rates

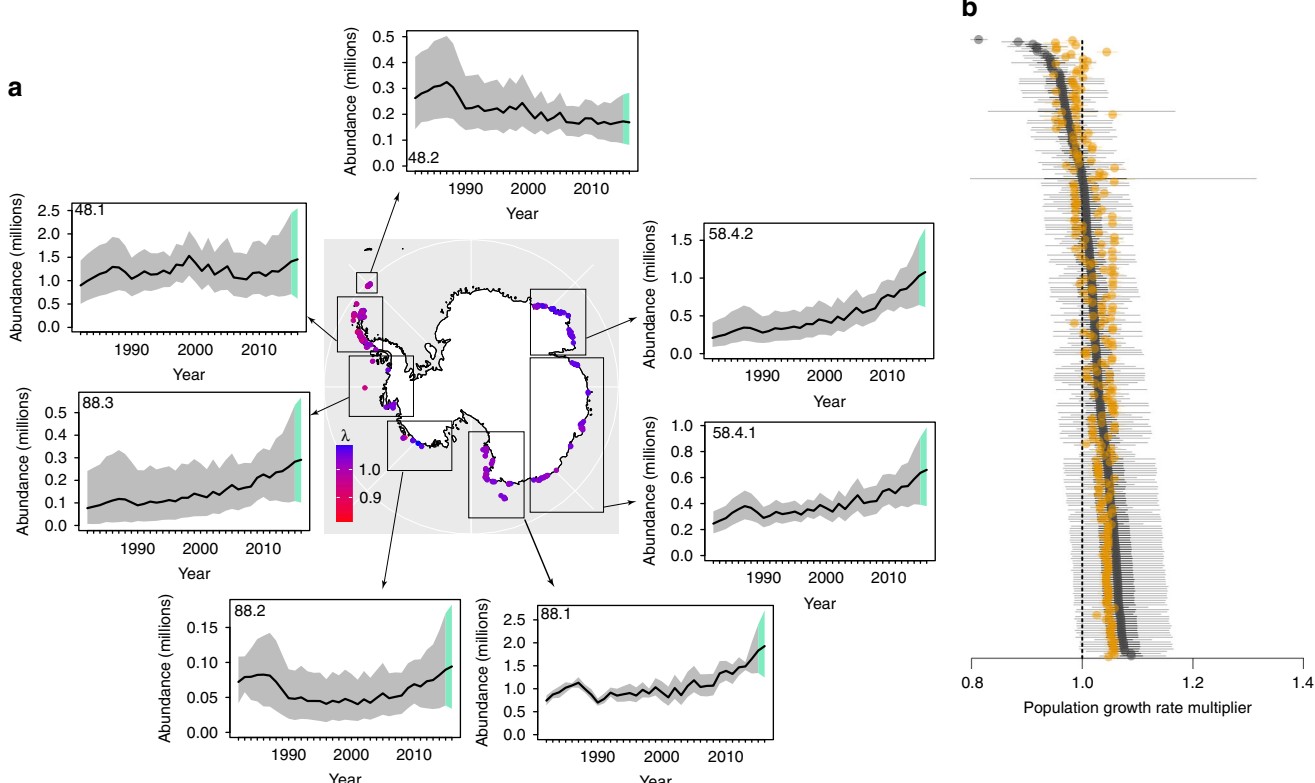

**Fig. 2** Spatially aggregated population dynamics. **a** Map (at *center*) displaying the posterior medians of the average actual population growth rate multipliers for all 267 Adélie colonies. For each site, this was computed as the geometric mean of the ratios of abundance in year *y*+1 to year *y* for all years the site was occupied between 1982 and 2015. The insets show Adélie abundance aggregated by CCAMLR subarea. The *gray* (1982–2015) and *green* (2016) *shaded areas* represent the 90% highest posterior density credible intervals; *black lines* are the posterior medians. Note that 2016 is beyond the end of our time series; all abundance estimates from 2016 reflect population forecasts from the model. **b** Average actual (*black circles* and *lines*) and predicted (*orange circles* and *lines*) population growth rate multipliers for all 267 Adélie colonies, ordered by magnitude. *Thick lines* represent the 50% equal-tailed credible intervals, *thin lines* represent the 95% equal-tailed credible intervals, and *circles* are the posterior medians. The average actual population growth rate multipliers were computed as in **a**. For each site, the average predicted population growth rate multiplier was computed as the geometric mean of the predicted growth rates ($e^r$; see Eq. (6) in Supplementary Data 1) for all years and the site was occupied between 1982 and 2015

The average population growth rate multiplier (i.e., the geometric mean of annual changes in nest abundance across all years, a site was occupied from 1982 to 2015) was highly variable across sites within the Antarctic Peninsula region (CCAMLR subarea 48.1); aggregated abundance across all sites in this region showed extended periods of both increasing and decreasing abundance over the last three decades (Fig. 2a). We also find a long-term decline in abundance in the South Orkney Islands, following an initial period of increase in the early 1980s (Fig. 2a). In contrast, we found a marked and steady increase in abundance around the rest of the Antarctic continent, including both Eastern Antarctica and the Ross Sea (Fig. 2a). Though deriving from a dynamical model for population abundance rather than data on colony occupancy, these results closely mirror the results from Adélie habitat suitability models based on presence/absence data[28].

Year-specific stochastic variation ("year effects"; Fig. 3a), which dominated the other population growth rate predictors in explaining intrinsic rate of growth (Fig. 1b), displayed no trend in mean or variance and was largely uncorrelated with traditional climate variables thought to influence Southern Ocean environmental conditions (Supplementary Data 4). Plots of population growth rate $r = \log\left(N_{y+1} N_y^{-1}\right)$ vs. population size suggested negative density dependence for some of the largest colonies (Supplementary Data 5).

Process error was roughly equivalent in magnitude to an observation error of 50% (see Supplementary Data 1 for a description of the accuracy categories associated with Adélie chick/nest counts), and 73% (95% credible interval: 69%, 86%) of all Adélie counts had observation errors smaller than the process error derived from the Adélie model (Fig. 3d). Simulated time series using the posterior median of process error and a biologically reasonable range of Adélie population growth rates (Fig. 2b) demonstrate that single-year anomalies in breeding abundance are common and even multi-year periods of increase or decrease can be generated in the absence of trending environmental covariates (Fig. 4; Supplementary Data 6). Aggregating over multiple sites greatly attenuates the random component of the underlying growth rates and has the power to detect trends with just a few years of data (Fig. 4; Supplementary Data 7–10).

## Discussion

Our results provide the most complete understanding to date of how inter-annual stochastic variation in seabird population growth rates can interact with long-term trends in abundance. If the population dynamics of Adélie penguins are as highly variable

as our model suggests (either due to genuine random forcing or the complex interaction of many unmeasured, or even unmeasurable, influences), the null expectation for time series of Adélie abundance should not be one of stasis but of fluctuation. Sudden changes in abundance can arise through several different, but not mutually exclusive, processes that operate at different temporal lags, including variability in the percentage of breeding age adults that skip breeding, and variation in adult survival, juvenile recruitment, and breeding productivity. While studies on banded penguins have differentiated among these processes for a small number of studied populations[17, 29], it is much more common that the drivers of fluctuating abundance at the breeding site remain unknown at spatial scales relevant to feedback management.

It is interesting that observation error was smaller than process error in this system, even with 15% of the surveys in our data set coming from the lowest precision category (Fig. 3d; Supplementary Data 1). Methods for estimating the abundance of Adélie penguins over large spatial scales have been developed for both medium- and high-resolution satellite sensors (e.g., Landsat, Worldview). Although observation errors associated with satellite surveys are currently larger than for traditional ground surveys[21, 30], our analysis implies that even relatively imprecise measurements of abundance may significantly contribute to our understanding of system dynamics and that the effort required to obtain very precise counts of abundance at fewer sites may be better spent getting lower precision estimates at more sites. Our model suggests that while observations of any accuracy can influence our estimate of abundance (see e.g., Cape Pigeon Rocks in Supplementary Data 10), observations with ±50% observation error or better will reduce our uncertainty in the abundance of penguins nesting at a site in any given year (see e.g., Paulet Island in Supplementary Data 8).

A global model of population dynamics, particularly for a species inhabiting over 24,000 km of circumpolar coastline, inevitably leaves out drivers of population growth specific to individual colonies or regions. While winter "ice droughts" and extensive summer sea ice have been discussed in previous studies of Adélie penguin population dynamics[5, 31–36], the links between sea ice and Adélie population dynamics are so complex that region-specific models may be necessary. Polynya size, for example, has been convincingly linked to the dynamics of adjacent populations[27], but was excluded from our global model because polynyas do not form throughout the entire Adélie penguin breeding range. Fortunately, continued data collection and new technologies are rapidly increasing the length, and decreasing the patchiness, of available time series, and our global model of dynamics is easily modified to include more tailored covariates if the size of the data set is sufficient for parameter estimation.

Adélie penguins are known to skip breeding in some years, with rates of skipping likely tied to environmental conditions[5]. We were not able to include a non-breeding state in our model due to the paucity of data for most of the time series involved. More complete time series, such as those that may be made available in the future through the integration of remotely sensed population estimates and long-term capture–recapture data, will permit the estimation of age- or stage-structured models. While

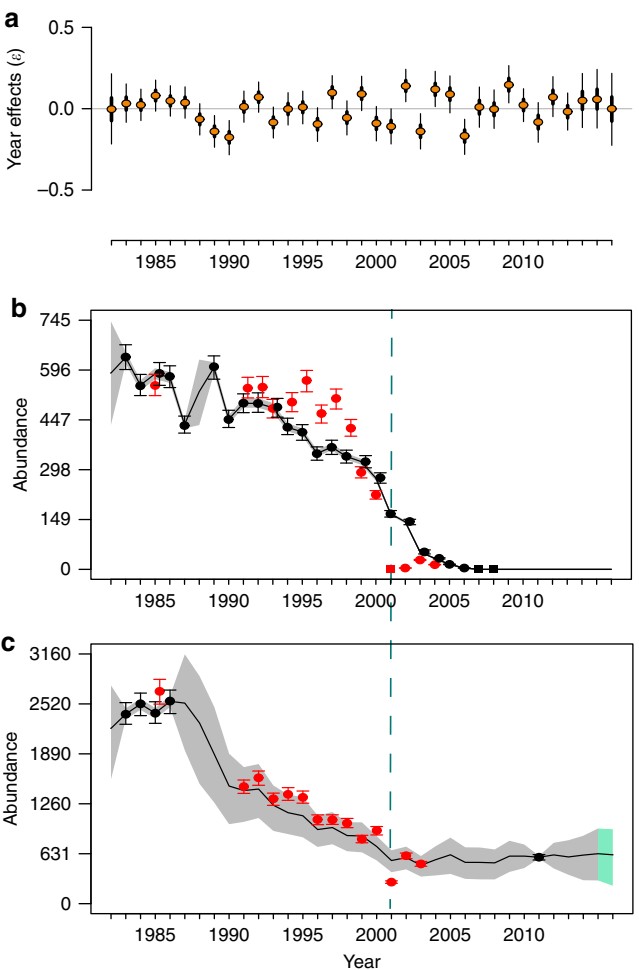

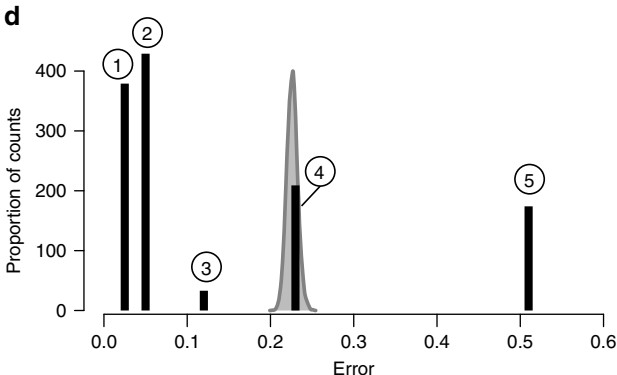

**Fig. 3** Impact of stochasticity on population dynamics. **a** Stochastic year effects ($\varepsilon_y$) on Adélie population growth rate ($r$). *Thick lines* represent the 50% equal-tailed credible intervals, *thin lines* represent the 95% equal-tailed credible intervals, and *circles* are the posterior medians. **b**, **c** Annual Adélie abundance at **b**, Litchfield Island, and **c** Humble Island, with Adélie nest (*black circles*) or chick (*red circles*) counts and the seasons where no nests (*black squares*) or chicks (*red squares*) were observed. The *gray* (1982–2015) and *green* (2016) *shaded areas* represent the 75% highest posterior density credible intervals; *black lines* are the posterior medians. The *error bars* represent the 90% highest posterior density credible intervals from the posterior predictive distributions for the nest or chick counts. Note that 2016 is beyond the end of our time series; all abundance estimates from 2016 reflect population forecasts from the model. **d** Histogram (*black bars*) of observation errors ($\sigma_{observation}$) associated with all counts included in the Adélie model overlaying a density plot (in *gray*) of the posterior distribution for the process error ($\sigma_{process}$) from the Adélie model. The numbers within *circles* over the *black bars* represent the accuracy code associated with each of these observation errors (Supplementary Data 1)

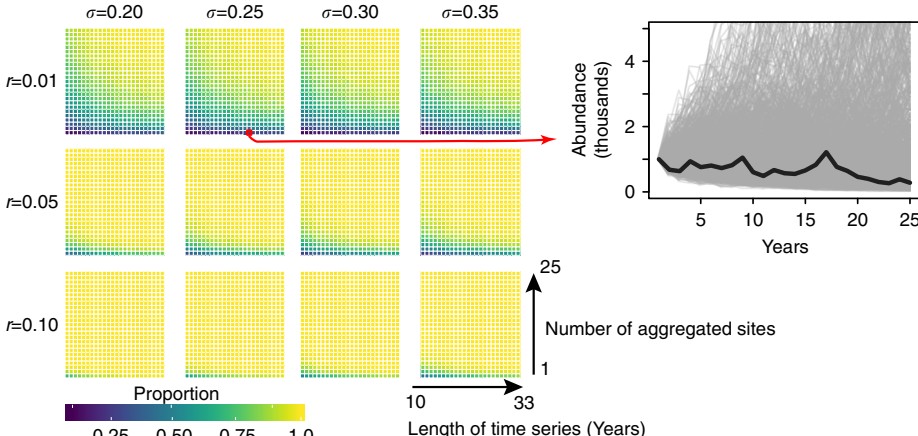

**Fig. 4** Recovery of population trends with process error. Heat maps showing the proportion of times the sign of the true intrinsic growth rate *r* was unambiguously recovered from time series of varying lengths (years) summed across varying numbers of sites. For each site, abundance was simulated using a simple exponential growth model with true intrinsic growth rate *r* as the median of a lognormal distribution whose scale parameter was *σ* beginning with an initial abundance of 1000 nests in year *y* = 1. The *inset* shows an ensemble of time series for one combination of population growth rate and process error, with a single trajectory highlighted in *black* for illustration

our model does accommodate spatiotemporal variation in breeding productivity and can therefore leverage information contained in chick or nest counts, age- or stage-structured models would illuminate whether process error derives mainly from skipped breeding or stochastic variation in demographic rates. However, any improvement in explanatory power over dynamical abundance models would require linking vital rates to explanatory covariates. Regardless of its cause, the role of process error has important consequences for penguin biologists looking to infer the cause of population changes. Given the scale of inter-annual variation in population abundances, time series of abundance can show extended periods of decline or increase due to stochastic variation alone. These anomalies should be treated with caution, as any attempt to correlate such changes with environmental conditions may draw false conclusions regarding the responsible driver.

Despite the influence of stochasticity in this system, our synthesis of all known Adélie penguin population data in Antarctica does permit the identification of several robust features. First, we find a convincing and widespread period of decline between 1987 and 1990, though we find that the previously suggested link between positive phases of the Southern Annular Mode[14] (SAM) and Adélie population declines is not robustly supported by the complete time series used in our model (Supplementary Data 4). The decline in the South Orkney Islands between 1987 and 2016 is consistent with previous work linking declines at individual sites in this region to bottom up factors resulting from reduced sea ice extent[23]. Commensurate with other studies (e.g., ref. [37]), we find that the population of Adélie penguins on the Antarctic Peninsula declined between 2000 and 2008, though we found an unexpected rebound in abundance starting in 2008. This regional increase in abundance may, in part, be driven by sites in the Marguerite Bay area, where Adélie penguins are stable or even increasing[22]. However, this increase may also reflect a cessation of regional warming on the Antarctic Peninsula since the late 1990s[38], which may benefit ice-dependent species like the Adélie penguin.

We find that while Eastern Antarctica appears to have been increasing steadily in abundance since at least 1982, the increasing abundance of Adélie penguins in the Ross Sea is more recent, beginning in 2002. While this is coincident with the increase in fishing pressure on Antarctic toothfish (*Dissostichus mawsoni*)[39], which compete with Adélie penguins for Antarctic silverfish (*Pleuragramma antarcticum*)[40, 41], other regions in Eastern Antarctica with lower toothfish catches also increased in abundance over the same period. Understanding the relative importance of toothfish or krill (*Euphausia* spp.) fishing vs. environmental effects would require much finer spatial resolution data on fishing than are currently available to the public, though our model could easily be extended to accommodate site-specific catch data should it become available in the future.

The 2001–2002 season has been identified as anomalous by several authors; unusual breeding, movement, or survivorship patterns have been variously attributed to the presence of a massive iceberg in the Ross Sea[17, 20, 42] and a series of atmospheric pressure anomalies that, among other things, created an unusually compact coastal ice barrier along the western Antarctic Peninsula[43, 44]. While this year was not extreme with respect to our model's "year effects" (Fig. 3a), we did find evidence for reduced breeding productivity in 2001 across multiple sites near Anvers Island including complete breeding failure (166 nests and 0 chicks) at Litchfield Island (Supplementary Data 7). Notably, 2001–2002 also had the lowest recorded breeding success for Adélie penguins in Terre Adélie, Eastern Antarctica[32].

Regardless of their cause, it appears that extreme events can tip small populations into a cycle of decline. For example, the already decreasing population at Litchfield Island declined rapidly to extinction following the 2001 breeding failure, while larger populations at neighboring sites such as Humble Island recovered (cf. Fig. 3b, c). This suggests that small populations may be vulnerable to even temporary disruptions caused by natural fluctuations in the environment. Although verified extinctions are such rare events that it is difficult to generalize, aerial predation by skuas (*Catharacta* spp.) can significantly decrease reproductive success in very small populations[45] (a component Allee effect). Unfortunately, we were unable to model the probability of a population going extinct as a function of population size (Eq. (12) in Supplementary Data 1). This is partially a statistical artifact of having few extinction events in the database but also reflects the species-agnostic nature of predator-driven Allee effects in this system. Small populations of Adélie penguins may be able to persist indefinitely if they breed in mixed colonies with other *Pygoscelis* spp. penguins, as often occurs where their ranges overlap on the Antarctic Peninsula. A multi-species dynamical

model that estimates the total abundance of all *Pygoscelis* penguins at a breeding colony is required to accurately estimate the strength of the Allee effects in population colonization and local extirpation.

We found a clear relationship between Adélie population size and growth rate (negative density dependence) at large population sizes (Supplementary Data 5). Previous work has found negative density dependence for Adélie populations in the Ross Sea[20], and our model suggests this pattern may extend to Adélie colonies located on the Antarctic Peninsula. This finding is unlikely to be driven by unfavorable sea ice conditions that may have occurred when Adélie populations were large, as predicted growth rates typically exceeded actual growth rates when population size was large (Supplementary Data 5). Because the carrying capacity of each site is different and few sites have enough data to allow for its estimation, we were not able to include a density-dependent term in this model.

Seabirds are considered good ecological indicators because they are thought to integrate conditions in a region surrounding the breeding colony and are far easier to survey than most of the prey species on which they depend. Adélie penguins have a long history of being used as an ecological indicator species and are one of the core components of CCAMLR's Ecosystem Monitoring Program (CEMP). Feedback management of marine resources, wherein changes in Adélie abundance are used to adjust catch limits or adaptively close certain fishing areas, requires that Adélie population dynamics can be used as a reliable indicator of marine ecosystem health. While this approach assumes that deterministic factors dominate stochastic influences, most models for Adélie penguin abundance have not explicitly included process error even though the combination of process and observation error is known to impact inference regarding key drivers[24], [46]. One of the major concerns regarding the use of seabird abundance as an indicator of marine ecosystem conditions is that abundance may change too slowly to be of value for the management of marine resources[47]. Our findings show that some seabirds may be characterized by the opposite problem, in that inter-annual fluctuations in abundance driven by unknown factors may be so large that it is difficult to assess trends in either the population status of the species or the "health" of the marine resources on which they depend.

In sum, we are left with an important question: Are Adélie penguins (or, in fact, any animal exhibiting this level of stochastic variability) too "noisy" to be used for feedback management? Our model and associated simulations show how significant shifts in abundance at individual colonies, even changes in abundance persisting over several consecutive years, can arise by the accumulation of process noise alone, and that the drivers of such changes may be difficult to establish. For all but the most extreme growth rates, even 20 years of data may be insufficient to reliably detect a trend at a single site (Fig. 4; Supplementary Data 6). These results suggest that individual breeding colonies may be too noisy to be of value for feedback management because the amount of time that would be required to measure a real trend may be far too long (Fig. 4). However, aggregating abundance across larger spatial management areas increases the signal to noise ratio for inferring trends in abundance and provides a spatially courser, but less noisy, depiction of abundance dynamics more likely to reflect underlying environmental conditions (Fig. 4). "Noisy" time series, such as those considered in this analysis, appear with regularity in ecology, and have attracted considerable theoretical attention among population ecologists (e.g., refs. [48–51]). In an applied context, however, a focus on the small fraction of total variability that can be explained may oversell what can be "read in the tea leaves" of individual time series. This can lead to a proliferation of post hoc explanations for

anomalies that undermine long-term management objectives. Fortunately, Bayesian approaches naturally facilitate the imputation of missing data, which are required for spatial aggregation of time series, and provide a statistically robust foundation on which to assess biologically relevant ecosystem variables.

## Methods

**Population dynamics model**. Using a hierarchical Bayesian population dynamics model, parameterized by all the publicly available data on Adélie penguin abundance and distribution since the 1982/1983 austral summer and accompanying sea ice data extracted from satellite imagery over the same period, we estimate a site-specific mean population growth rate and an abundance in each year since 1982 for each of Antarctica's 267 known Adélie breeding populations (see Supplementary Data 1–2 for a comprehensive explanation of the Adélie model, and details regarding modeling fitting, checking, and validation). Population growth rate was modeled as a linear function of peak sea ice across the last five winters ($y$−4 to $y$), which we included as a surrogate for krill recruitment; summer sea ice in year $y$−4, which relates to colony access in the birth year for new recruits to the breeding population; and a random effect for year shared by all the colonies. Random site and CCAMLR subarea effects to capture site or management unit differences not otherwise captured by spatial covariates were not supported by the data and were not, therefore, included in the final model. Likewise, covariates related to krill catch (reported by CCAMLR Secretariat's Data Centre[52]) were not found to be strongly correlated with population growth rates and are also excluded from the final model. We modeled reproductive success (chicks per occupied nest) hierarchically, drawing breeding productivity values for each site by year combination from a truncated normal distribution that spans the possible range of penguin productivity. We relied on prior information regarding reproductive success[53] in order for the mean and variance of this distribution to be identifiable. Our Adélie model provides a mechanism to estimate the abundance of populations in each year regardless of whether there were census data for those years, which is critical to aggregating across sites to construct continuous time series of abundance at spatiotemporal scales relevant for feedback management (Supplementary Data 7–10). This is accomplished by predicting nest abundances in years with missing count data for each site, and, if necessary, hindcasting nest abundance recursively from the initial year of count data backwards to 1982 using the inverse of the exponential growth function (Supplementary Data 1).

We report process error from the Adélie model as the scale parameter from the lognormal distribution governing true abundance (Eq. (5) in Supplementary Data 1) and compare that process error to the observation errors associated with all nest and chick counts included in our model (Eqs. (1–2) in Supplementary Data 1). We used simulated time series to explore the consequences of stochastic forcing across a range of growth rates and process errors consistent with output from the Adélie model for time series of breeding abundance of varying lengths aggregated across varying numbers of sites (Supplementary Data 6).

**Code availability**. All model code and associated documentation are available in Supplementary Data 1–3.

**Data availability**. Additional analyses associated with this manuscript are provided in Supplementary Data 4–10. The Adélie abundance and sea ice concentration data that support the findings of this study are publicly available through the Mapping Application for Penguin Populations and Projected Dynamics website (MAPPPD)[54], http://www.penguinmap.com. Supplementary Data 3 provides code to query MAPPPD's underlying PostgreSQL database, which is publicly available on Amazon RDS, and format the query results as model input.

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

## Acknowledgements

H.J.L., C.C.-C., G.H., C.Y., and K.T.S. gratefully acknowledge funding provided by US National Aeronautics and Space Administration Award No. NNX14AC32G and U.S. National Science Foundation Office of Polar Programs Award No. NSF/OPP-1255058. S.J., L.L., M.M.H., Y.L., and R.J. gratefully acknowledge funding provided by US National Aeronautics and Space Administration Award No. NNX14AH74G. H.J.L., C.Y., S.J., Y.L., and R.J. gratefully acknowledge funding provided by U.S. National Science Foundation Office of Polar Programs Award No. NSF/PLR-1341548. S.J. gratefully acknowledges support from the Dalio Explore Fund. We would like to thank Julienne Stroeve and Garrett Campbell from the National Snow and Ice Data Center (Boulder, CO, USA) for their guidance and discussions about sea ice dynamics.

## Author contributions

H.J.L., S.J., and C.Y. conceived the project; C.C.-C. and H.J.L. wrote the manuscript; C.C.-C., H.J.L., and K.T.S. performed the modeling; C.C.-C., G.H., P.M., and H.J.L. developed the penguin database; L.L., M.M.H., Y.L., R.J., S.J., and C.Y. provided covariate information; All authors provided comments on the manuscript and interpreted results.

## Additional information

**Competing interests:** The authors declare no competing financial interests.

