## [Peer Review File · Nature Communications]

Reviewers' comments:

Reviewer #1 (Remarks to the Author):

This is an interesting piece of work that considers issues of general interest to management bodies responsible for Antarctic fisheries.

A number of the ecological and management issues considered by the manuscript have previously been considered by CCAMLR, albeit using an older and simpler analysis framework. These analyses and considerations are reported in the CCAMLR Scientific Committee Report for 2003, with more complete details in Appendix D of the report from the Working Group on Ecosystem Monitoring and Management. The report is publically available (<https://www.ccamlr.org/en/system/files/e-sc-xxii.pdf>).

The following terms of reference for the WG-EMM work were established (SC-CAMLR-XX, Annex 4, paragraphs 5.16 and 5.17):

- (i) Are the nature and use of the existing CEMP data still appropriate for addressing the original objectives?
- (ii) Do these objectives remain appropriate and/or sufficient?
- (iii) Are additional data available which should be incorporated in CEMP or be used in conjunction with CEMP data?
- (iv) Can useful management advice be derived from CEMP or be used in conjunction with CEMP data?

The report recognised that Bayesian approaches could be used as an alternative to the power analysis undertaken.

An important conclusion from the report was that at current harvesting levels it was unlikely that the existing design of CEMP, with the data available to it, would be sufficient to distinguish between ecosystem changes due to harvesting of commercial species and changes due to environmental variability, whether physical or biological. The report further noted that with the existing design of CEMP it may never be possible to distinguish between these different and potentially confounding causal factors.

Although the report only considered natural biological and environmental variability, a further source of variability that will probably also be important in any revised analysis of Adélie monitoring data are data on recovering cetacean populations.

Each of these issues should be considered in this paper. Taking this previous work into account will ensure the submitted manuscript will be much more comprehensive, building on the previous deliberations by CCAMLR.

Reviewer #2 (Remarks to the Author):

General Comments

Strengths

Adélie Penguins are a keystone Antarctic species that are used extensively as an indicator species for management purposes, especially by the Antarctic Treaty System through bodies like CCAMLR. By modelling a time series of Adélie Penguin population data from the entire Antarctic continent over

multiple decades, Che-Castaldo et al. provide important and novel insights into the drivers of population change, and in doing so, clarify the primary sources of error in Adélie penguin population dynamics.

The paper is generally well-written, the analyses are well thought out, sophisticated, and comprehensive. The results are backed up by full disclosure of both the data used and the code used to run the analyses. Extensive Supplementary materials provide clear and important background data and analytical information.

The authors make a compelling case for the importance of stochastic processes, and/or unmeasurable variables (captured by including the term 'year' as a random effect in the model) as the most important sources of error in understanding the population trends of Adélie penguins. The authors also make the important claim that aggregating population data over multiple sites has the power to detect trends with much less data, and their results support this claim.

Aspects of the manuscript that require attention

At times the paper uses too much jargon. While terms like 'process error' and 'average population growth multiplier' are reasonably well known in the field of population dynamics, they are probably overused and should, at the very least, be more clearly explained the first time they are used

The paper is occasionally let down by speculation that does not appear to have a strong evidence base (see specific comments for more details).

One issue that should be addressed further is the absence of any spatial structure in the models. The inclusion of site as a random effect may have gone some way to address this, but the authors show that as a covariate it had little influence and negatively affected convergence. This I find a little strange, as many of the stochastic or unmeasurable processes ostensibly represented by the 'year' term would likely be better represented by a spatial term. To address the lack of spatial structure, and to try and account for spatial autocorrelation in the data, I would like to see the inclusion of a spatial term in the model. Something like a specific bivariate term for location.

The authors should also consider testing the inclusion of the CCAMLR statistical sub-area as random effect. While only representing very broad spatial structure, this might help to account for the autocorrelation in the foraging areas used by individuals nesting in the same general area.

It's not entirely clear in the paper why the authors didn't include some of the environmental covariates outlined in the analyses presented in Supplementary Materials 4, directly in the model, rather than looking at the cross-correlation of these with the 'year' term. There may be a good reason for this, but if so it needs to be explicitly outlined.

Finally, although it may be beyond the scope of this paper, it would be very interesting to run these models at a regional scale (eg for each of the CCAMLR statistical sub-areas) to clarify if different covariates were having different effects across the different regions. Such models could use regional specific covariates and likely provide much more accurate insights into drivers of population change, while still benefiting from the multiple site advantages of the models developed here.

Specific comments

L34 Suggest the relationship between the two errors would be better described if 'dominate' was replaced by 'exceeds' in this sentence

L38 Although the term "process error" is reasonably well known in the field of population monitoring, I think the underlying causes leading to this type of error need to be made more explicit for general readership. Therefore, I suggest changing the last sentence of the Abstract to read "...driven by stochastic, or unmeasurable processes".

L74-75 To the best of my knowledge there is no empirical data that links krill harvest to with krill density. I think this is too speculative to include here, and it needs to be deleted or qualified more.

L77-78 I'm not convinced by the relatively ad-hoc inclusion of this covariate in the model. I would prefer to see the analyses report with the best model (or at least have the results of the best model reported and compared)

L79 'average population growth multiplier' needs to be better explained at first use for more general readership

L87 Looking at Fig 1b, I wouldn't say this effect 'dominated ' the other predictors, but I do think it clearly exhibits the 'strongest effect'

L89-91 Was any attempt made to include some of the these covariates directly in the model ? This would seem to be a more direct way of clarifying the influence of these parameters on population dynamics.

L152-153 Again, a better explanation is needed as to why environmental variables (like the SAM mentioned here) were only tested against the year effect and not included directly in the model.

L170-172 This is very speculative, and not backed by empirical data. Suggest reword or delete

L192-208 I found this paragraph to be poorly structured, and quite confusing. There is speculation about the effects of skuas (but again, no empirical data), and then more unfounded speculation about the benefits of breeding in mixed species colonies. Needs to be reworded to convey a clear message.

L273-274 While I understand the exclusion of the site random effect based on the information provided in Supp 1, I still think more attention needs to be paid to the impact of spatial structure in the population data. Also see suggestion above in General Comments

L285 'publically' is more commonly spelled 'publicly'

Supplementary Materials

Supp 1 Additional assumptions

Converting the adult counts to nest counts is likely to result in considerable overestimation of numbers. It would be useful to see a breakdown of how many of the counts were actual nest counts, compared to adult counts and chick counts.

Recommendation

Accept with revision

Reviewer #3 (Remarks to the Author):

General comments

In this manuscript, the authors collate publically available Adelie penguin breeding abundance data from around Antarctica, use the data to parameterize a new population model, use the parameter estimates to simulate time series at 268 breeding sites around the continent, and make conclusions about the implications of process and observation error on power to detect change in relation to spatial scale. The approach is novel and the goal is ambitious. This is an interesting paper that has the potential to address an important issue in resource monitoring and management, which, in this case, could be relevant for feedback management of the krill fishery in the Southern Ocean.

While addressing power in relation to process and observation error for monitoring and management is the main thesis, much of the paper focuses on other issues which distract from its main purpose and in many instances is unconvincing. The paper purports to make 'findings' about population change around Antarctica when in fact many of these changes have already been reported in the scientific literature. It is disingenuous to collate much of the data presented in regional studies and claim them as a 'new findings', which in any case are peripheral to the main thesis. Furthermore, the tone of the manuscript confuses real results with predictions from the model, and this needs to be clarified throughout. There is discussion of issues such as anomalous events and density dependence which are interesting from an ecological perspective but are largely peripheral to the main thesis of monitoring and management. The manuscript would be improved by having a clearer focus of what the paper is about rather than attempting to be so broad in its approach

While I commend the ambitiousness and scope of the thesis, I think the authors over-reach in relation to the supporting data and stated results from the population model. First, the authors' claim to have collated all publically available data is not true and should not be made (see some examples below). Second, while the authors emphasize upfront in the discussion that a pan-Antarctic population model does not account for regional variation and hence is likely to omit important drivers and explanations of population change, they proceed with doing precisely this. This issue is exaggerated by the fact that 98.5% the data used to parameterize the model come from just two regions. This is very likely to lead to biased conclusions when applying a global model to regions with few data. Indeed, some of the pan-Antarctic model predictions which they claim are 'robust and widespread' are not supported by publically available data from the data-poor regions which the authors have overlooked. A more appropriate approach would be to apply the model to just the regions where sufficient data exist and include region specifically as a factor in the model, thereby allowing the issue of region-specific dynamics to be addressed and avoiding over-reach in their conclusions specifically for regions with little data. Thirdly, the population model is complex and only poorly explained despite descriptions in many supplements. It is critical that a full explanation and thorough review of the model is made before it is used for management.

Specific comments

Management context

Lines 228-231. Feedback management of the krill fishery is an area of active work within CCAMLR and studies that can contribute to an effective feedback management system are important. The authors characterize the application of feedback management by CCAMLR in terms of a single species (the Adelie penguin) and a single response variable (abundance) and in doing so give the impression that feedback management of the krill fishery is a relatively simple, agreed and established practise that this work is directly relevant to. In fact, feedback management is still under active discussion in CCAMLR, there has been no agreement on how it will be applied in the future, predators in addition to Adelie penguins may be used (eg fur seals, flying seabirds) and some feedback management models under consideration do not even use abundance as a response variable. It is understandable that the

authors may not be fully aware of the current discussions within CCAMLR as they are not closely involved in them, but if their work is to be useful under this management context, it is important that their portrayal of CCAMLR's work is accurate and this is not currently the case.

Lines 231-235. Further to the above comment, while I agree that most published models that focus on Antarctic predators do not explicitly address process and observation error, these issues have been long recognized, researched and discussed in CCAMLR. The lack of focus in the published literature is probably because most previous work has been undertaken and aimed at academic biologists and ecologists and to answer ecological questions rather than management questions. I welcome a wider focus on management that papers like this can offer, but again stress that it is important that CCAMLR's work is portrayed accurately.

Data

Lines 61-62. The authors claim that they use all publically available data on Adelie penguin distribution and abundance since 1982/83 is not true and the claim should not be made. Some examples of data that are publically available but are not included in the MAPPPD database are provided in comments below. There are others.

Supplement 1, VII Additional assumptions, assumption 2. This assumption states that adult counts were treated as equivalent to nest counts in relation to bias (i.e. 1:1 with nest counts as stated) but are less precise than nest counts. The authors claim to address this issue by assigning a higher (less precise) observation error to adult counts than was attributable to the count itself. This treatment confuses the concepts of bias (uni-directional error) and precision (bi-directional error). As the authors state, counts of adults will usually include non-breeders, so there will almost always be more adults present, and never less, than occupied nests (i.e. the error is unidirectional). Changing the precision will not address this kind of error. Also, the ratio of adults to nests changes across the breeding season so the extent of bias in assuming a 1:1 relationship will vary according to the date of a count. The bias associated with this treatment of adult counts will propagate through to biased estimates of population growth, which may affect the model predictions. Either adult counts should be standardized to address bias, or not included in the analysis.

Supplements 8-10. While the collated data are indeed pan-Antarctic in spatial extent (but not comprehensive, see comments above and below), the data are very strongly dominated by the extensive time series data in Areas 48.1 (Antarctic Peninsula) and 88.1 (Ross Sea). The authors clearly state this spatial bias in Section V of Supplement 1 (Abundance process), and I commend them on their openness in this regard. However, this does raise an important issue that is at the heart of this paper whose main thesis is an analysis of pan-Antarctic data with pan-Antarctic predictions and conclusions for resource and conservation management. Without meaning to be flippant, if I word the issue something like in the spirit of the wording of the title, it is: how big a data gap is too big for reliable prediction and management? The data that are used to parameterize the model that is used to make pan-Antarctic predictions is year-to-year abundance estimates and, based on the figures in Supplements 8-10, by my count the breakdown of these data across the three main CCAMLR regions is:

Area 48: 182 yearly transitions, (46.3% of all pan-Antarctic transitions)

Area 88: 206, (52.2%)

Area 58: 6, (1.5%)

Clearly any parameter estimates and subsequent predictions about Area 58 will be driven almost entirely by processes that occur in Areas 48 and 88, and as these three regions differ significantly in their ecologies, there is a strong risk that population dynamics and 'predictions from a global' model will be biased in any region.

The model structure, predictions and findings

General comments

This is a complex model that requires a more detailed explanation than is available in the text and supplementary material to be thoroughly reviewed. This includes better explanation as to how the errors in model predictions are generated, how the time series are seeded for initial population size, whether forward and backward forecasting is used, and how robust the extrapolations could be at sites for which there is so little data. As far as I am aware, there is not even an attempt at validating the model. For a model that the authors hope to be used in management, this is a serious limitation.

Supplement 1, Section IV, Environmental covariates

The expectation of sea-ice cover as an important driver of Adelie populations is reasonable given the results of several other studies, but the decision to summarize sea-ice data within 500 km of breeding sites for both winter and summer sea-ice needs justification. In particular, how consistent are the environmental covariates used here with previous studies showing environment-response relationships, and is a fixed distance of influence at all colonies around Antarctica for both summer and winter justified by published foraging studies. It would be useful to refer to such studies.

Lines 71-74.

The location of the reference to figure 1 gives the impression in the text that figure 1 shows the ecological (blocking access) and demographic (reduced breeding success and recruitment) consequences of summer sea-ice, when it only shows the magnitude of the association relative to other factors. The appropriate place for a reference to figure 1 is immediately after the mention of the association (ie after ...'four years prior')

Lines 79-86, 172-175. Pan-Antarctic trends in abundance over the last 30 years.

The authors claim to make 'findings' about changes and trends in populations in the Antarctic Peninsula, Ross Sea, and east Antarctica. These are not new findings, as the same conclusions have been recently published in three excellent regional studies (Lynch et al 2012, Lyver et al 2014, Southwell et al 2015 respectively) and a pan-Antarctic study (Lynch and LaRue 2014), using original data. Repeating the same conclusions by simply collating previously published data, or filling in the gaps in time series using model predictions, in my view does not constitute a new finding. It is interesting that none of the regional papers have been cited in the paper in this context, and one was not cited at all. In addition to the change and trend results having been previously published, assessing trends is not central to the main theme of process and observation error and therefore is not pertinent to the paper's thesis. This comes back to clarifying the purpose of the current submission.

Lines 50-54. Robust predictions?

The authors argue their analysis identifies a number of robust features including 'a convincing and widespread period of decline between 1987-1990'.....and.....'a pan-Antarctic drop in population growth in 2001'. Given that a global model is used and therefore does not allow for variation between regions or sites, some pan-Antarctic predictions and conclusions would be expected. However, given the strong spatial bias in the data used to parameterize the model, such pan-Antarctic conclusions may be an artefact of this data bias. There is evidence from uncited papers that this may be the case or it simply reflects similar environmental fluctuations for environmental covariates included in the model. Kato and Ropert-Coudert (2006), for example, show trends in Adelie abundance from 1980 to the early 2000's at ten sites along the Soya coast of East Antarctica, none of which concur with the above conclusions. Similarly, results from Bechervaise Island in East Antarctica in Clark et al (2003) show population growth was higher, not lowest, in 2001/02 than any of the previous 11 breeding seasons. The authors also claim a complete breeding failure at the only site with publically available nest and chick count data in that year as supporting evidence for their finding. However, the nest and chick

counts in Clark et al (2003), which are publically available but neither included in the MAPPPD database nor cited in the paper, show breeding success in this year was not unusually low in 2001/02 but was in fact the second highest in any year from 1990/91 to 2001/02. These publically available but uncited results do not concur with the two predicted continent-wide events of population decline and breeding failure and instead support the proposition that the pan-Antarctic predictions are biased by the imbalanced data currently available.

Lines 202-206. Following up further on the breeding failure at Litchfield Island in 2001/02, the authors argue that the proposed continent-wide anomalous event in 2001/02 may have tipped the small population at Litchfield Island into a rapid decline and extinction and refer to figure 3 in support of this position. However, figure 3 actually shows the population declining at an increasing rate since 1985, and suggests the driving force was likely a long term process operating over three decades rather than as a consequence of a single anomalous year.

Figure 2 and Supplements 8-10. Predicted time series

There are several things I find puzzling in the predicted time series in Figure 2 and Supplements 8-10. A comment above raises the fact that almost all the data used to parameterize the model came from areas 48.1 and 88.1. One might expect then, that the predicted regional-scale time series for these regions would be more certain, and hence have smaller confidence envelopes, than for other regions. So I am comfortable with the time series for area 88.1 in Figure 2 having a smaller confidence envelope than any other region. But why would area 48.1, which contributed almost half the data, have a wider confidence envelope than areas 58.4.1 and 58.4.2, which together contributed only 1.5% of the data? In fact, why do areas 58.4.1 and 58.4.2 have such small error bounds when many of the time series have only one or two data points available upon which to contribute to the model? Another thing that has me confused is what is used to seed these time series when initial data are scarce. The time series have a scale that is clearly meant to represent the real number of penguins present, but many time series at specific sites have only one actual data point associated with them that may be anywhere along the time series and is often towards the end. Are the actual data points used to seed the time series predictions, and if so are time series back- and/or forward-projected from these points? This is another aspect of the model that is so poorly explained that renders it difficult to review.

A third issue I can find no explanation for is how or if the actual data and the predicted time series in Supplements 8-10 interact or link. Sometimes the predicted time series converges or constrains to mimic the actual data completely, and sometimes it does not. As an example, Chistoffersen Island (and several other sites) has one actual data point and a predicted time series with a large confidence envelope that is constrained to be smaller than the confidence interval for the only actual data point. Why and how does this happen?

Lines 102-103, 129-132. Aggregating sites

The authors argue that demographic processes such as intermittent breeding, survival and juvenile recruitment will contribute to process or random error, and predict from the model that aggregating abundance data over sites attenuates the random component of growth. It is not clear to me how this issue is addressed or accommodated in the model when it does not include a site factor to allow for site-specific effects.

Lines 100-104, 138-147, and 242-261. Power to detect change

Drawing the threads together from these various sections of the paper, the authors argue that aggregating abundance across sites reduces random or process noise (see comment above on whether the model can realistically predict this), hence methods that can estimate abundance at large scales will produce less process-noisy time series and have more power to detect change, and new

satellite methods provide a means to doing this, even if abundance estimates are less precise than other methods applied at smaller scales. No-one would doubt that any method that can estimate populations at large scales frequently, accurately and precisely is highly desirable, regardless of the issue of process error. The issue is, are there methods currently available, or can new methods be developed, that meet all the specific requirements of the power predictions (scale, frequency, accuracy, precision). Satellite methods offer this potential, but I don't think the field is at the stage where it is feasible to detect change within just the few years alluded to in lines 102-103 and these threads of text could be taken to imply. It is ambiguous whether this is a theoretical prediction or a prediction that has in mind currently available satellite methods, and this should be clarified. The evidence I am aware of indicates that satellite methods are still too imprecise to be able to detect change within a few years, even at aggregated sites. For example, most of the abundance estimates for multiple aggregated sites derived from high resolution satellite imagery estimates in Lynch and LaRue 2014 have precisions of $\pm 60\%$ which is far too large to detect anything other than a massive change within a few years with high confidence.

References

- Clark et al (2003) Demographic characteristics of the Adélie penguin population on Bechervaise Island after 12 years of study. CCAMLR Sci.10: 53-74.
- Kato A, Ropert-Coudert Y (2006) Rapid increase in Adélie penguin populations in the Lützow-Holm Bay area since the mid 1990s. Polar Biosci 20: 55-62.
- Lynch HJ, Naveen R, Trathan PN, Fagan WF (2012) Spatially integrated assessment reveals widespread changes in penguin populations on the Antarctic Peninsula. Ecology 93: 1367-1377.
- Lyver POB, Barron M, Barton KJ, Ainley DG, Pollard A, Gordon S, et al. (2014) Trends in the breeding population of Adélie penguins in the Ross Sea, 1981-2012: a coincidence of climate and resource extraction effects. PLoS ONE 9: e91188.
- Lynch HJ, LaRue MA (2014) First global census of the Adélie penguin. The Auk 131: 457-466.
- Southwell, C., Emmerson, L., McKinlay, J., Takahashi, A., Kato, A., Barbraud, C., Delord, K. and Weimerskirch, H. (2015) Spatially extensive standardized surveys reveal widespread, multi-decadal increase in East Antarctic Adélie penguin populations. PLoS ONE 10 (10): e0139877.

Reviewer #4 (Remarks to the Author):

Overall comments

I was asked to check the modeling and the coding, both of which I found OK with nothing amiss. In general, the manuscript is well written and the figures illustrate relevant results that are interesting and worthy of publication. The Discussion, though, could be improved in the following ways:

- The Discussion would read better if its opening paragraph began with the text that is currently on L 120. The opening paragraph could begin with:

Our results provide the best understanding of how interannual stochastic variation in seabird 122 population growth rates interact with/influence(?) long-term trends in abundance. If the populations of Adélie penguins are as variableAND CONTINUE TILL THE END OF THE PARAGRAPH

- L 109-120 could either be deleted, or follow the above as the second paragraph

- Start a new paragraph on L 192: the new paragraph will be the extinctions para

- The Discussion is long. One section that could go is L 209-217

- You can start this cropped paragraph at L 218 with something like:

The level of process error in a population has important consequences ...TILL END OF THE PARAGRAPH". You will end up with a short paragraph, but that's OK.

Running the models via someone like Andy Royle (USGS, Patuxent) would be a good idea. Andy is a busy person, so he may agree if its only the models to review. -Another section that could go is L 224-

241. It is interesting, but not necessary. Moreover, I am not sure if the statement on L 233-34 "most models for Adélie penguin abundance have not included process error" is correct. Don't older (non-Bayesian) population models include process error, but not observation error? Don't hierarchical (Bayesian) models improve on older system process models by allowing the observation process to be modeled separately from the system process, which then allows the introduction of an observation error? - L 252-261 is unnecessary, i.e., delete from: 'Noisy' time series ...TILL END OF PARAGRAPH." IF L 252-261 ARE OMITTED, THEN 1-2 FRESH SENTENCES ARE NEEDED TO END THE DISCUSSION.

Minor edit

In Supplementary 2, given that three chains is mentioned a couple of times in the text, it would be best to change `n.chains = 1` in the `jags.model` call to `n.chains = 3`.

Reviewer 1

1. *A number of the ecological and management issues considered by the manuscript have previously been considered by CCAMLR, albeit using an older and simpler analysis framework. These analyses and considerations are reported in the CCAMLR Scientific Committee Report for 2003, with more complete details in Appendix D of the report from the Working Group on Ecosystem Monitoring and Management. The report is publicly available (<https://www.ccamlr.org/en/system/files/e-sc-xxii.pdf>). The following terms of reference for the WG-EMM work were established (SC-CAMLR-XX, Annex 4, paragraphs 5.16 and 5.17).*
 - (a) *Are the nature and use of the existing CEMP data still appropriate for addressing the original objectives?*
 - (b) *Do these objectives remain appropriate and/or sufficient?*
 - (c) *Are additional data available which should be incorporated in CEMP or be used in conjunction with CEMP data?*
 - (d) *Can useful management advice be derived from CEMP or be used in conjunction with CEMP data?*

The report recognized that Bayesian approaches could be used as an alternative to the power analysis undertaken. An important conclusion from the report was that at current harvesting levels it was unlikely that the existing design of CEMP, with the data available to it, would be sufficient to distinguish between ecosystem changes due to harvesting of commercial species and changes due to environmental variability, whether physical or biological. The report further noted that with the existing design of CEMP it may never be possible to distinguish between these different and potentially confounding causal factors. Although the report only considered natural biological and environmental variability, a further source of variability that will probably also be important in any revised analysis of Adelie monitoring data are data on recovering cetacean populations. Each of these issues should be considered in this paper. Taking this previous work into account will ensure the submitted manuscript will be much more comprehensive, building on the previous deliberations by CCAMLR.

REPLY: We appreciate the Reviewer bringing the 2003 CCAMLR Scientific Committee Report to our attention, and we have read through Appendix D carefully. We certainly do not mean to imply that these issues are new, only that they are perhaps even more significant than is recognized in the Antarctic community (which is far larger than the CCAMLR community and its working groups) and even more so, the broader ecological community. We have added some text to highlight CCAMLR's work in this area (Lines 60–70), though a complete review of CCAMLR's past work in this area would be both beyond the scope of our paper and of only limited interest to most readers. We would like to note that this working group report specifically mentions that “identifying the sources of variability in CEMP indices can illustrate whether improvements can be made by alternative levels of data aggregation”. Our

findings address this issue directly and suggest that aggregating population data of several sites can increase the signal to noise ratio for management of Southern Ocean resources.

Reviewer 2

1. *At times the paper uses too much jargon. While terms like “process error” and “average population growth multiplier” are reasonably well known in the field of population dynamics, they are probably overused and should, at the very least, be more clearly explained the first time they are used.*

REPLY: We have added some explanatory text at the first mention of “process error” (Line 56) and a mathematical explanation of “average population growth multiplier” (Line 87) to aid the reader. In addition we have clarified the role of process and observation error in Supplement 1 and added text to the Results (Lines 104–107) and included a new figure contrasting the magnitude of these errors in our model (Fig. 3d).

2. *One issue that should be addressed further is the absence of any spatial structure in the models. The inclusion of site as a random effect may have gone some way to address this, but the authors show that as a covariate it had little influence and negatively affected convergence. This I find a little strange, as many of the stochastic or unmeasurable processes ostensibly represented by the “year” term would likely be better represented by a spatial term. To address the lack of spatial structure, and to try and account for spatial autocorrelation in the data, I would like to see the inclusion of a spatial term in the model. Something like a specific bivariate term for location.*

REPLY: In our model development, we included a random site effect but the variation associated with site (beyond what was already captured by the covariates, which are spatially structured and likely capture a lot of the standing covariation in population growth rates) was virtually zero (see Fig. R1 below) and did not include these effects in the manuscript. In the resubmitted model, we include a spatial component by modeling site \times year breeding productivity hierarchically (in the original submission breeding productivity was a fixed value across all sites and years). While breeding productivity does vary substantially by site and year (Fig. S7-6), we feel this likely reflects environment conditions (such as precipitation during the breeding season) as opposed to a strict site and/or year effects. We were unable to include random site effects on intrinsic rate of growth in this revised model due to convergence issues. Given our experience trying to fit a model with both spatial and temporal variation, a full spatial covariate model is unlikely to be estimable given the sparseness of the data at this point. As the dataset grows over time, we certainly see the value in exploring spatial covariance models that might allow for better prediction of poorly sampled sites in proximity to well sampled sites, as doing so may improve predictive performance. It is worth noting some spatial covariance is likely driven by over-winter behavior, which can be modeled better once we have more information on overwinter habitat use. Such

data are likely to emerge over time with greater focus on tracking penguins foraging over the non-breeding period.

3. *The authors should also consider testing the inclusion of the CCAMLR statistical sub-area as random effect. While only representing very broad spatial structure, this might help to account for the autocorrelation in the foraging areas used by individuals nesting in the same general area.*

REPLY: We did consider including a random effects for CCAMLR sub-area but this random effect explained virtually none of the variation in growth rate (see Fig. R2 below) and we did not include it in the manuscript. We note that the lack of a random effect for CCAMLR sub-region was not surprising, both because the covariates we believe are important for penguin biology have been included directly and also because any remaining drivers are likely to follow climatological divisions, not the management divisions reflected in the CCAMLR sub-areas. We added text to the Methods (Lines 286–288) explaining why we did not include random effects for CCAMLR sub-area.

4. *It's not entirely clear in the paper why the authors didn't include some of the environmental covariates outlined in the analyses presented in Supplementary Materials 4, directly in the model, rather than looking at the cross-correlation of these with the 'year' term. There may be a good reason for this, but if so it needs to be explicitly outlined.*

REPLY: We included in this model all the covariates with an *a priori* biological hypothesis. Our discussion in Supplement 4 was to try and post hoc understand what might be driving the year effect we observed. On a practical level, this was done as a separate step because the model is so computationally intense that it was not possible to fit a very large number of models, as would be required when exploring such a comprehensive number of climate indices. In terms of applied management, it is worth noting that including climate-related indices as predictors may actually increase predictive uncertainty because the climate indices themselves are themselves predictions. Lastly, the cross correlations in Supplement 4 shows that penguin responses to climate is spatially explicit and, hence, will involve regional models where regions will be defined climatologically (as opposed to using traditional management areas). We see Supplement 4 as a starting point for this work which is outside the scope of this paper.

5. *Finally, although it may be beyond the scope of this paper, it would be very interesting to run these models at a regional scale (eg for each of the CCAMLR statistical sub-areas) to clarify if different covariates were having different effects across the different regions. Such models could use regional specific covariates and likely provide much more accurate insights into drivers of population change, while still benefiting from the multiple site advantages of the models developed here.*

REPLY: We completely agree that finer scale sub-models may ultimately provide the best predictive performance. Models developed for individual sites can be quite

detailed but do not scale to larger spatial units, as are required for spatial management of marine resources. Some regions are particularly data sparse, and a full optimization of spatial groupings for model development was beyond the scope of this analysis. One of the reasons we have made such an extraordinary effort to make our model available to other researchers is that we hope those with regional expertise will be able to adapt this model in a way that captured finer scale drivers of penguin population growth. We would like to caution, however, that there are relatively few covariates that are actually measured at the site level, and with few exceptions, the penguin community is restricted to covariates that can be measured by satellites.

6. L34: *Suggest the relationship between the two errors would be better described if “dominate” was replaced by “exceeds” in this sentence*

REPLY: We have made this suggested edit (Line 34).

7. L38: *Although the term “process error” is reasonably well known in the field of population monitoring, I think the underlying causes leading to this type of error need to be made more explicit for general readership. Therefore, I suggest changing the changing the last sentence of the Abstract to read “driven by stochastic, or unmeasurable processes.”*

REPLY: We have made this suggested edit (Line 38). In addition, see response to Query #1.

8. L74–75: *To the best of my knowledge there is no empirical data that links krill harvest with krill density. I think this is too speculative to include here, and it needs to be deleted or qualified more.*

REPLY: Krill is no longer included in the model presented in this paper, both because it explains very little of the variation in population growth rate, and also because we do not have data on either krill catch or krill densities with the spatial resolution that would be required. We added text to the Methods (Lines 288–291) explaining why we did not include the krill covariate in our model.

9. L77–78: *I’m not convinced by the relatively ad-hoc inclusion of this covariate in the model. I would prefer to see the analyses report with the best model (or at least have the results of the best model reported and compared)*

REPLY: We had originally included krill because we had an *a priori* interest in krill as a covariate, but our revised manuscript now presents the model without krill catch since, as noted in the original manuscript, the krill catch data are at a very coarse spatial scale and thus do not explain a significant fraction of the variation in internal growth rates. In addition, see response to Query #8.

10. L79: *“average population growth multiplier” needs to be better explained at first use for more general readership*

REPLY: As noted above, we have added a short explanation of “average population growth multiplier” at its first mention (Lines 87–88).

11. L87: *Looking at Fig 1b, I wouldn't say this effect “dominated” the other predictors, but I do think it clearly exhibits the “strongest effect”*

REPLY: See response above noting our change to “exceeded” rather than “dominated”. In addition, see response to Query #1.

12. L89–91: *Was any attempt made to include some of these covariates directly in the model? This would seem to be a more direct way of clarifying the influence of these parameters on population dynamics.*

REPLY: The only covariate that was included in Cimino et al. that was not included in our model was sea surface temperature, which was used in that case as a proxy for more ultimate measures of biological productivity. In this case, we were hoping to use satellite-derived estimates of chlorophyll-a, however the satellite record for chlorophyll-a does not extend back beyond the early 1990s and we have elected to use a single model for the entire time series. While sea surface temperature is likely a good proxy for spatial variation in occupancy because it covaries with other environmental factors, we do not think that sea surface temperature is a good proxy for inter-annual variation in population growth rate because its effects on krill density are likely occurring at much slower time scales.

13. L152–153: *Again, a better explanation is needed as to why environmental variables (like the SAM mentioned here) were only tested against the year effect and not included directly in the model.*

REPLY: See response to Query #4.

14. L170–172: *This is very speculative, and not backed by empirical data. Suggest reword or delete*

REPLY: It is uncontroversial to suggest that ice dependent species would benefit from a cessation in climate warming in this region. The cessation of warming on the Antarctic Peninsula drives from previously published work, and thereby deserves mention in this context. As this comment occurs in the Discussion, we feel comfortable advancing this explanation as one of many possible explanations for the unexpected increase in abundance in subarea 48.1.

15. L192–208: *I found this paragraph to be poorly structured, and quite confusing. There is speculation about the effects of skuas (but again, no empirical data), and then more unfounded speculation about the benefits of breeding in mixed species colonies. Needs to be reworded to convey a clear message.*

REPLY: We have modified this paragraph to address the Reviewer’s concerns, and have added a citation about the role of skuas and reproductive success (Lines 211–222).

16. L273–274: *While I understand the exclusion of the site random effect based on the information provided in Supp 1, I still think more attention needs to be paid to the impact of spatial structure in the population data. Also see suggestion above in General Comments*

REPLY: Like the Reviewer, we also expected that a random effect for site would reduce process error; however, this was not the case, likely because the effect of site was adequately captured by the spatial covariates. See also response above to Query #2. Greater study on the spatial covariance relationships is certainly of interest, but given the computational challenges of running the model, would benefit from a priori information on foraging and/or migration, the details of which were outside the scope of our paper.

17. L285: *“publically” is more commonly spelled “publicly”*

REPLY: We have changed this throughout.

Reviewer 3

1. *While addressing power in relation to process and observation error for monitoring and management is the main thesis, much of the paper focuses on other issues which distract from its main purpose and in many instances is unconvincing. The paper purports to make “findings” about population change around Antarctica when in fact many of these changes have already been reported in the scientific literature. It is disingenuous to collate much of the data presented in regional studies and claim them as a “new findings”, which in any case are peripheral to the main thesis. Furthermore, the tone of the manuscript confuses real results with predictions from the model, and this needs to be clarified throughout. There is discussion of issues such as anomalous events and density dependence which are interesting from an ecological perspective but are largely peripheral to the main thesis of monitoring and management. The manuscript would be improved by having a clearer focus of what the paper is about rather than attempting to be so broad in its approach.*

REPLY: We agree that many excellent analyses have been published in the last few years focused on aspects of Adélie penguin population dynamics that are incorporated

in our analysis here (and cited in the Introduction), and we have made every effort to refer to those earlier studies in our paper. However, some of these results have been presented in local or regional-scale analyses and here are putting them in the context of continent-wide changes. Our paper is focused on the difficulties that arise when significant inter-annual variation in abundance makes it difficult to establish the drivers of change in time series of abundance. We also show that spatial aggregation provide one means of overcoming this issue to extract meaningful (from a management perspective) information on short time scales. We are not sure what is meant by the comment “the tone of the manuscript confuses real results with predictions from the model”, though in our revision of the manuscript we have made sure that every mention of the data or the model clearly distinguishes data from model predictions. It is worth noting that are analysis provides a unifying framework that will be beneficial to the community moving forward to understand not only what is common among these regions but also what distinguishes them from one another. Finally, it is very important to recognize that our model takes a sparse matrix of observed data and creates a complete matrix of predicted abundance, as is *required* if we are to understand population dynamics at larger spatial scales. There is no other model that has been developed to date that allows for regional aggregation. Such spatial aggregation is the only way to know whether declines at one site are compensated for at other sites, and is critical for management of marine resources at CCAMLR-relevant spatial scales. While we expect (and hope) that the details of the “best-available” model will improve over time, our approach provides a unifying framework on which future progress in this area hinges.

2. *While I commend the ambitiousness and scope of the thesis, I think the authors over-reach in relation to the supporting data and stated results from the population model. First, the authors claim to have collated all publically available data is not true and should not be made (see some examples below).*

REPLY: While we made every effort to uncover all the data that were available, we are delighted that the review process has uncovered some additional sources of data. We have added all of the data suggested by the Reviewers that we were able to obtain. The most significant additions were from a paper by Kato and colleagues on survey data in Eastern Antarctica, though we note that the raw data were actually not available in the manuscript (and thus were not in the public domain) and had to be requested from the authors. Additional data sources were identified, and the data requested for our analysis, but in these other cases the data owners were unwilling to share those data and thus they could not be included. We would like to note that in many cases, it may appear as though data or key references are missing from our database, however in many cases this is due to one of three reasons: (1) It may be that the site name that we use is not the same as that used in the original source, since it was often the case that several names were used for a single penguin population. We have created a standard list of names and site codes and so it may appear that a site is missing when in fact it has just been renamed in the process of creating a

standard data set. This is one of the key outputs from this work, since the site names have been a constant source of confusion over the decades (see Southwell et al. 2017). (2) There are many cases where a single census count was listed in several references, and so it might appear that a reference is “missing” when in fact it is simply that the data contained in that reference was attributed to a different publication. It is often the case that a later publication, which included a comprehensive time series including previously published data, was used as the reference of record, rather than the earlier publication. (3) There are several key references which included abundance data at smaller or larger spatial scales than individual breeding populations (which was our spatial unit since it is the most directly tied to demographic processes). While we have contacted the original authors to try and obtain data at comparable spatial scales, these data were not always available and thus could not be included in our analysis.

3. *Second, while the authors emphasize upfront in the discussion that a pan-Antarctic population model does not account for regional variation and hence is likely to omit important drivers and explanations of population change, they proceed with doing precisely this. This issue is exaggerated by the fact that 98.5% the data used to parameterize the model come from just two regions. This is very likely to lead to biased conclusions when applying a global model to regions with few data. Indeed, some of the pan-Antarctic model predictions which they claim are 'robust and widespread' are not supported by publically available data from the data-poor regions which the authors have overlooked. A more appropriate approach would be to apply the model to just the regions where sufficient data exist and include region specifically as a factor in the model, thereby allowing the issue of region-specific dynamics to be addressed and avoiding over-reach in their conclusions specifically for regions with little data.*

REPLY: We are not sure how Reviewer #3 determined that “98.5% the data used to parameterize the model come from just two regions”. Of the 1,223 counts used in our model, 38% of them come from the Antarctic Peninsula (464 counts), 36% from western Antarctica (438 counts), and 26% from eastern Antarctica (321 counts). While we recognize the limitations imposed by data scarcity in several of the CCAMLR regions, we think it is better to use the available data in a comprehensive model rather than simply excluding some regions, the latter of which presents no useful information to the management community. In this regard, we note that 75% of all Adélie penguins breed in either 48.1 or 88.1, so these data rich sites also happen to be the ones with the majority of the penguins; as such, their considerable influence on the model is neither unexpected nor inappropriate. Also, Adélie penguins have limited physiological plasticity and so we think that data rich areas can be used to infer something about likely responses to environmental conditions in areas with less data. We would also like to note that when we went back and added all the data that was suggested by the manuscript Reviewers, it did not change our model results nor the inference derived from our model. We cannot address the concerns that our model is contradicted by other datasets because the Reviewer does not provide enough specific information for us to confirm or deny the charge, and with the additional data added since the

manuscript's initial review, we are at a loss as to what other data may be still missing. Finally, it is important to keep in mind that the continental scale of our analysis provides a straightforward and transparent means to add additional data when and if they become available, and that the code that has been provided with the Supplementary Materials can be used by others in the community to update or modify the model as needed.

4. *Thirdly, the population model is complex and only poorly explained despite descriptions in many supplements. It is critical that a full explanation and thorough review of the model is made before it is used for management.*

REPLY: Our Supplementary Materials are comprehensive in terms of providing both code and the text to understand the model assumptions. We have revised Supplement 1 to be even more clear with regards to the source and role of observation error in the model and the ways in which we quantify and compare process and observation errors. In the Methods, we have added text that points the reader to the supplementary materials when necessary. We believe we have gone above and beyond in this analysis to provide a comprehensive verbal description of the model, along with the mathematical details to satisfy other modelers in the field. Note that we have divided the Supplementary Materials into 10 different documents, each of which is hyperlinked between the table of contents and the relevant sections to aid readability. Of these 10 Supplements, all of the model details and code are contained in the first three Supplements, which provide all the information needed to not only understand the model but also reproduce the model results. All of the Reviewers were provided with the tools to reproduce our analysis *in its entirety*. In fact, before submission, we tested our Supplements with other researchers to ensure that our results can be fully reproduced. There is simply nothing else that could have been done. As to the issue of model complexity, our model is a simple exponential growth model with three covariates on population growth rate. It is the data that are complicated, and accommodating data complexity drives the bulk of our model code. Unfortunately, there is nothing that we can do to make the data less complicated without overly simplifying the data available or discarding unreasonable amounts of data in this already data poor system.

5. *Lines 228-231. Feedback management of the krill fishery is an area of active work within CCAMLR and studies that can contribute to an effective feedback management system are important. The authors characterize the application of feedback management by CCAMLR in terms of a single species (the Adélie penguin) and a single response variable (abundance) and in doing so give the impression that feedback management of the krill fishery is a relatively simple, agreed and established practise that this work is directly relevant to. In fact, feedback management is still under active discussion in CCAMLR, there has been no agreement on how it will be applied in the future, predators in addition to Adélie penguins may be used (eg fur seals, flying seabirds) and some feedback management models under consideration do not even use abundance as a response variable. It is understandable that the authors may not be fully aware of*

the current discussions within CCAMLR as they are not closely involved in them, but if their work is to be useful under this management context, it is important that their portrayal of CCAMLR's work is accurate and this is not currently the case.

REPLY: We do not mean to imply that feedback management within CCAMLR solely hinges on the Adélie penguin, and in fact we know that many species are being used as indicators of change in the Antarctic by CCAMLR. However, the Adélie penguin has been for decades one of the key species being monitored within CEMP. Moreover, several aspects of our analysis extend to other CCAMLR target species. For one, the overarching role of process error and the complexity introduced by process error in management, extend to other target species as well. There is no reason to expect that Adélie penguins are unique in this regard. Second, our modeling approach (particularly using a Bayesian model to predict abundance for years with missing data, and thereby allowing for spatial aggregation of patchy time series) is extensible to other CCAMLR target species and thus our paper has broader impacts within the CCAMLR community.

- 6. Lines 231-235. Further to the above comment, while I agree that most published models that focus on Antarctic predators do not explicitly address process and observation error, these issues have been long recognized, researched and discussed in CCAMLR. The lack of focus in the published literature is probably because most previous work has been undertaken and aimed at academic biologists and ecologists and to answer ecological questions rather than management questions. I welcome a wider focus on management that papers like this can offer, but again stress that it is important that CCAMLR's work is portrayed accurately.*

REPLY: We agree with the Reviewer, and are resigned to the fact that in a manuscript of this length it would not be feasible to adequately describe CCAMLR's discussions with respect to these issues, noting that CCAMLR's work stretches back many decades. We address the Reviewer's concern in detail in our response to Reviewer #1 and have edited our introduction to more directly address CCAMLR's discussions of these issues (including the role of process error in CEMP indices) and to note that our model provides a solution to CCAMLR's suggestion that spatiotemporal aggregation of CEMP indices may be required (see citation provided by Reviewer #1 which we have included in the manuscript).

- 7. Lines 61-62. The authors claim that they use all publically available data on Adélie penguin distribution and abundance since 1982/83 is not true and the claim should not be made. Some examples of data that are publically available but are not included in the MAPPD database are provided in comments below. There are others.*

REPLY: We have added all additional sources of data that were highlighted by Reviewer #3. See also response to Query #2.

8. *Supplement 1, VII Additional assumptions, assumption 2. This assumption states that adult counts were treated as equivalent to nest counts in relation to bias (i.e. 1:1 with nest counts as stated) but are less precise than nest counts. The authors claim to address this issue by assigning a higher (less precise) observation error to adult counts than was attributable to the count itself. This treatment confuses the concepts of bias (uni-directional error) and precision (bi-directional error). As the authors state, counts of adults will usually include non-breeders, so there will almost always be more adults present, and never less, than occupied nests (i.e. the error is unidirectional). Changing the precision will not address this kind of error. Also, the ratio of adults to nests changes across the breeding season so the extent of bias in assuming a 1:1 relationship will vary according to the date of a count. The bias associated with this treatment of adult counts will propagate through to biased estimates of population growth, which may affect the model predictions. Either adult counts should be standardized to address bias, or not included in the analysis.*

REPLY: The Reviewer is correct that there may also be a bias inherent to converting adult counts to nest counts, however we disagree that this error is always unidirectional. For example, adult counts may be reported when surveys are done very early in the season (prior to nesting), when nests have not yet been established and the number of adults may actually be smaller than the ultimate number of nests. Notwithstanding the issue of bias vs. precision in this conversion, the adult counts in our updated database come overwhelmingly (86%) from time series (published by Kato and others) comprised only of adult counts, which means that while our estimate of actual abundance may be biased upwards, it will not bias our inference regarding the dynamics at these sites, nor the relationship between population growth rate and environmental drivers. It is worth noting that several of the authors of this paper (Lynch, Jenouvrier, Youngflesh, McDowall) are involved in another project in which UAVs were used to survey both adults and nest counts at the peak of nesting, and we find that a count of adults are only 5-20% larger than a count of occupied nests. While this is no guarantee that the adult-to-nest bias inherent to the Kato et al. counts are similarly small, it does suggest that an appropriately timed adult count is closely related to the number of nesting pairs and so the impact on our estimate of breeding pairs in that region will be minimal.

9. *Supplements 8-10. While the collated data are indeed pan-Antarctic in spatial extent (but not comprehensive, see comments above and below), the data are very strongly dominated by the extensive time series data in Areas 48.1 (Antarctic Peninsula) and 88.1 (Ross Sea). The authors clearly state this spatial bias in Section V of Supplement 1 (Abundance process), and I commend them on their openness in this regard. However, this does raise an important issue that is at the heart of this paper whose main thesis is an analysis of pan-Antarctic data with pan-Antarctic predictions and conclusions for resource and conservation management. Without meaning to be flippant, if I word the issue something like in the spirit of the wording of the title, it is: how big a data gap is too big for reliable prediction and management? The data that are used to*

parameterize the model that is used to make pan-Antarctic predictions is year-to-year abundance estimates and, based on the figures in Supplements 8-10, by my count the breakdown of these data across the three main CCAMLR regions is: Area 48: 182 yearly transitions, (46.3% of all pan-Antarctic transitions) Area 88: 206, (52.2%), Area 58: 6, (1.5%). Clearly any parameter estimates and subsequent predictions about Area 58 will be driven almost entirely by processes that occur in Areas 48 and 88, and as these three regions differ significantly in their ecologies, there is a strong risk that population dynamics and predictions from a global model will be biased in any region.

REPLY: We do not disagree, which is why we were clear in our manuscript to highlight the heterogeneity in the data across the different regions of Antarctica. However, as noted above, 75% of Adelie penguins do come from subareas 48.1 and 88.1, and we do not see a biological reason why the fundamental role of process noise would be different in the less well sampled regions. We appreciate very much the Reviewer's comments in this regard, but do not believe that these issues in any way affect our results or discussion. We would like to emphasize that it is not within our power to make the available data more complete, we are simply trying to accommodate the data scarcity in the most appropriate way to understand the population dynamics of this important indicator species.

10. *This is a complex model that requires a more detailed explanation than is available in the text and supplementary material to be thoroughly reviewed. This includes better explanation as to how the errors in model predictions are generated, how the time series are seeded for initial population size, whether forward and backward forecasting is used, and how robust the extrapolations could be at sites for which there is so little data. As far as I am aware, there is not even an attempt at validating the model. For a model that the authors hope to be used in management, this is a serious limitation.*

REPLY: In terms of describing the details of the model, the information requested is included in the Supplementary Materials 1, and we have added additional information to Supplementary Materials 1 and to the paper's Methods to clarify the issues raised by the Reviewer. In particular, we have added to the Supplementary Materials more detail on the process by which models are fit, how accuracy counts are translated into observation error, and how the model is used to hindcast back from that first count year to the missing data at the beginning of the time series. In response to Reviewer #3's concerns about validation, we have added an additional cross-validation check on our model. Here we have randomly removed 10% of the data and have compared our model-predicted abundance against the actual abundance. We find that 96% of our predicted values fall within the 95th percentile confidence intervals for the predictive distributions, indicating good predictive performance. We have added text in the Methods pointing readers to Supplement 1 for the details regarding the cross-validation.

11. *Supplement 1, Section IV, Environmental covariates. The expectation of sea-ice cover as an important driver of Adelie populations is reasonable given the results of sev-*

eral other studies, but the decision to summarize sea-ice data within 500 km of breeding sites for both winter and summer sea-ice needs justification. In particular, how consistent are the environmental covariates used here with previous studies showing environment-response relationships, and is a fixed distance of influence at all colonies around Antarctica for both summer and winter justified by published foraging studies. It would be useful to refer to such studies.

REPLY: Our choice of spatial scale reflects a balance between what is biologically meaningful and what can be meaningfully extracted from the satellite-based record. For example, it may be that fine-scale sea ice dynamics (e.g., cracks in the ice, etc.) may be biologically important, but the available satellite imagery-derived sea ice products do not provide information at that spatial scale. Likewise, satellite-derived sea ice products can be unreliable very close to shore. Our use of 500 km as a spatial scale for the sea ice covariates reflects the best available information we can get that relates to the biological processes relevant to Adélie penguin foraging and prey availability. Earlier in our analysis, we did examine correlations between population growth rate and sea ice metrics at various spatial scales, and 500 km was chosen as the optimal balance between capturing local scale influences and, at the same time, having sea ice covariates that were not too noisy due to variations in the sea ice product (in other words, averaging over more pixels increased the signal to noise ratio). The 500 km radius covariate appears to capture some aspect of environmental conditions more relevant to the biology of the process, which may reflect a true biological mechanism or, as discussion above, something about the satellite products themselves.

12. *Lines 71-74. The location of the reference to figure 1 gives the impression in the text that figure 1 shows the ecological (blocking access) and demographic (reduced breeding success and recruitment) consequences of summer sea-ice, when it only shows the magnitude of the association relative to other factors. The appropriate place for a reference to figure 1 is immediately after the mention of the association (ie after... four years prior).*

REPLY: As suggested we have changed the reference to Figure 1; it is now on Line 86.

13. *Lines 79-86, 172-175. Pan-Antarctic trends in abundance over the last 30 years. The authors claim to make findings about changes and trends in populations in the Antarctic Peninsula, Ross Sea, and east Antarctica. These are not new findings, as the same conclusions have been recently published in three excellent regional studies (Lynch et al 2012, Lyver et al 2014, Southwell et al 2015 respectively) and a pan-Antarctic study (Lynch and LaRue 2014), using original data. Repeating the same conclusions by simply collating previously published data, or filling in the gaps in time series using model predictions, in my view does not constitute a new finding. It is interesting that none of the regional papers have been cited in the paper in this context, and one was not cited at all. In addition to the change and trend results having been previously*

published, assessing trends is not central to the main theme of process and observation error and therefore is not pertinent to the paper's thesis. This comes back to clarifying the purpose of the current submission.

REPLY: We have edited the introduction (e.g., Lines 50–79) to clarify the focus of our paper and to reiterate that many of the regional scale trends have been highlighted in previously published studies.

14. *Lines 50-54. Robust predictions? The authors argue their analysis identifies a number of robust features including a convincing and widespread period of decline between 1987-1990.....and... a pan-Antarctic drop in population growth in 2001. Given that a global model is used and therefore does not allow for variation between regions or sites, some pan-Antarctic predictions and conclusions would be expected. However, given the strong spatial bias in the data used to parameterize the model, such pan-Antarctic conclusions may be an artefact of this data bias. There is evidence from uncited papers that this may be the case or it simply reflects similar environmental fluctuations for environmental covariates included in the model. Kato and Ropert-Coudert (2006), for example, show trends in Adelie abundance from 1980 to the early 2000's at ten sites along the Soya coast of East Antarctica, none of which concur with the above conclusions. Similarly, results from Bechervaise Island in East Antarctica in Clark et al (2003) show population growth was higher, not lowest, in 2001/02 than any of the previous 11 breeding seasons. The authors also claim a complete breeding failure at the only site with publically available nest and chick count data in that year as supporting evidence for their finding. However, the nest and chick counts in Clark et al (2003), which are publically available but neither included in the MAPPPD database nor cited in the paper, show breeding success in this year was not unusually low in 2001/02 but was in fact the second highest in any year from 1990/91 to 2001/02. These publically available but uncited results do not concur with the two predicted continent-wide events of population decline and breeding failure and instead support the proposition that the pan-Antarctic predictions are biased by the imbalanced data currently available.*

REPLY: We are confused as to why the Reviewer says that sites and regions cannot vary, because they do vary according to the spatially structured covariate information on sea ice and krill. Where sites have high precision data (accuracy 1 and/or accuracy 2 count - see Tables 1 and 2 in Supplement 1), the model will in fact track the empirical data very closely, and so sites will predicted to increase or decrease as appropriate given the information contained in the available data. To clarify, at the risk of being redundant with earlier statements, the only sense in which sites and regions do not “vary” is that we have not included a random site or region intercept on top of the covariates, the latter of which has a direct mechanistic connection to trends in abundance. (As noted above, we did explore models with a random effect for site, and this random effect did not capture any of the residual variation in population growth rates and was, for that reason, not included in the final model.) We have noted elsewhere in this letter that we now include data from Kato and Ropert-Coudert (2006) and the

other papers mentioned by Reviewer #3 (noting, again, the difficulties of including data in cases where it is not available at the correct spatial scale). As to the issue of 2001/02, after amending our model to allow breeding productivity to vary across space and time (see revised Methods and Supplement1), the strong “year effect” seen in 2001 is now decomposed into a smaller “year effect” and a reduced breeding success (which is now separately accounted for in our revised model with the addition of a year-and-site-varying reproductive success). We have updated the Results and Discussion (Lines 196–205) to highlight the fact that although this year was not extreme with respect to other “year effects” (Fig. 3a), we did find evidence for reduced breeding productivity in 2001 across multiple sites near Anvers Island. Also, we have now included the data from Clarke et al. (2003) and since this does provide another data point for chick production in 2001/02, we have amended our statement about this in the manuscript (Lines 200–203).

15. *Lines 202-206. Following up further on the breeding failure at Litchfield Island in 2001/02, the authors argue that the proposed continent-wide anomalous event in 2001/02 may have tipped the small population at Litchfield Island into a rapid decline and extinction and refer to figure 3 in support of this position. However, figure 3 actually shows the population declining at an increasing rate since 1985, and suggests the driving force was likely a long term process operating over three decades rather than as a consequence of a single anomalous year.*

REPLY: We agree with the Reviewer, in fact our manuscript explicitly states that “interannual stochastic variation in seabird population growth rates can interact with long-term trends in abundance” and that “small populations may be vulnerable to even temporary disruptions”. We have used Humble Island instead of Cape Crozier, to illustrate this point as Litchfield Island and Humble Island both suffered breeding failures in 2001/02 and were geographically near one another. In Figure 3, we are highlighting that an extreme event will impact a declining population differently than a stable or increasing population, and that while both Litchfield Island and Humble Island saw a sudden decline in breeding abundance in 2001/02, the larger and growing Humble Island population quickly bounced back while the Litchfield Island population quickly blinked out. We have revised the Discussion text in Lines 206–211 to reflect these changes. Specifically, we added the phrase “already decreasing population” to Line 207 to reflect the Reviewer’s point that the Litchfield population was already in decline.

16. *Figure 2 and Supplements 8-10. Predicted time series. There are several things I find puzzling in the predicted time series in Figure 2 and Supplements 8-10. A comment above raises the fact that almost all the data used to parameterize the model came from areas 48.1 and 88.1. One might expect then, that the predicted regional-scale time series for these regions would be more certain, and hence have smaller confidence envelopes, than for other regions. So I am comfortable with the time series for area 88.1 in Figure 2 having a smaller confidence envelope than any other region. But why would*

area 48.1, which contributed almost half the data, have a wider confidence envelope than areas 58.4.1 and 58.4.2, which together contributed only 1.5% of the data? In fact, why do areas 58.4.1 and 58.4.2 have such small error bounds when many of the time series have only one or two data points available upon which to contribute to the model?

REPLY: The Reviewer has highlighted an interesting emergent property of our results. The confidence intervals for a region are driven primarily by the number of sites within that region that are poorly sampled, and because 48.1 has so many sites within it, there remain many poorly sampled sites within 48.1 that are driving the relatively large uncertainty in total abundance. Furthermore, much of the survey work in 48.1 occurred in two periods (see Supplement 8, Figure S8-2), one early in the analysis period and then in the most recent period. The compounding dynamical uncertainty in the intervening period leads to ballooning confidence intervals (see Clark and Bjørnstad 2004 *Ecology*) in the middle of the time series. Note that if we had the same amount of data distributed more uniformly through time, we would not have this ballooning uncertainty at those sites and the regional abundance would not be so uncertain. Contrast this case with 88.1, where we had a moderate number of sites but the data are more evenly distributed across the data matrix. As a result, the confidence intervals are quite narrow. In sum, the confidence intervals on regional abundance are a function of how missing data are distributed in time, and how large the poorly sampled sites are relative to those with better sampling. We think this is an important feature of the model, because it properly captures the uncertainty stemming from these poorly sampled sites when looking at abundance over larger spatial scales. This information is itself useful, as it allows managers to explore various sampling scenarios and to allocate sampling effort to reduce regional uncertainty in total penguin abundance. An exploration of this was beyond the scope of this manuscript but similar analyses have been completed in the past (see Lynch et al. 2012 *Antarctic Science*).

17. *Another thing that has me confused is what is used to seed these time series when initial data are scarce.*

REPLY: The easiest way to think about the model in this respect is to think of each model as beginning with the first data point that exists in the time series, and being fit moving forward from that starting point. Once the model is fit (for all time series, which now form a ragged array due to the different starting points), each time series is simulated in reverse starting from the first data point that exists to hindcast the initial portion of the time series that was missing. For this reason, the confidence intervals for the time series flare out in the early years of the time series due to the hind casting procedure, especially where the data start relatively late in the time series. We have added text in the Methods regarding how hindcasting was used to backfill abundance for each site in the years prior to the initial count and updated Supplement 1 with the details regarding hindcasting.

18. *The time series have a scale that is clearly meant to represent the real number of penguins present, but many time series at specific sites have only one actual data point associated with them that may be anywhere along the time series and is often towards the end. Are the actual data points used to seed the time series predictions, and if so are time series back- and/or forward-projected from these points? This is another aspect of the model that is so poorly explained that renders it difficult to review.*

REPLY: See response to Query #17 above, and the revised description of the hind-casting procedure in Supplement 1.

19. *A third issue I can find no explanation for is how or if the actual data and the predicted time series in Supplements 8-10 interact or link. Sometimes the predicted time series converges or constrains to mimic the actual data completely, and sometimes it does not. As an example, Chistoffersen Island (and several other sites) has one actual data point and a predicted time series with a large confidence envelope that is constrained to be smaller than the confidence interval for the only actual data point. Why and how does this happen?*

REPLY: The Reviewer raises several points that can be confusing and we try to clarify this here. In regards to “I can find no explanation for is how or if the actual data and the predicted time series in Supplements 8-10 interact or link.”: In our model, the time series of true abundance (which we assume is what Reviewer #3 is referring to as the predicted time series) is conditional on the actual data since the observation model and biological model are linked through the true abundance z as described in Supplement 1. On all site-level figures (Chistoffersen Island included), the gray credible intervals represents the uncertainty in true abundance z^* from the model. The black bars are the posterior predictive intervals for the counts themselves, meaning the interval in which the model expects to find an nest or chick given the associated observation error (which is provided by the observers and not estimated). These intervals reflect uncertainty in the true nest abundance and/or uncertainty in the breeding productivity (in the case of chick counts), as well as observer measurement error, and hence these intervals are larger than the credible intervals around true abundance. We have added text to Supplement 1 to reflect this point. Lastly, we used 75% credible intervals on true abundance and 90% credible intervals on the predicted counts in the site level plots so the changes in abundance over time and the true counts were visually easy to follow. Given the high process error, uncertainty balloons out in years of missing data, and for some sites can compress the time series against the x-axis when we used matching 90% credible intervals on true abundance.

20. *Lines 102-103, 129-132. Aggregating sites The authors argue that demographic processes such as intermittent breeding, survival and juvenile recruitment will contribute to process or random error, and predict from the model that aggregating abundance data over sites attenuates the random component of growth. It is not clear to me how*

this issue is addressed or accommodated in the model when it does not include a site factor to allow for site-specific effects.

REPLY: We are not entirely sure that we understand what the Reviewer is asking, but in this portion of the manuscript we are simply pointing out that some of the process error might be explained by the aforementioned demographic processes. It is important to note that while we tried a site-specific random effect and it did not explain a significant fraction of the process error, we cannot address whether a specific source of process error (such a skipped breeding) is best modeled by a random “site effect” since we do not have data on these additional demographic components. While we have not addressed skipped breeding, we mention it here in the Discussion since due diligence requires us to raise all the factors that may influence process error and that might be accommodated were additional data to be collected.

21. *Lines 100-104, 138-147, and 242-261. Power to detect change. Drawing the threads together from these various sections of the paper, the authors argue that aggregating abundance across sites reduces random or process noise (see comment above on whether the model can realistically predict this), hence methods that can estimate abundance at large scales will produce less process-noisy time series and have more power to detect change, and new satellite methods provide a means to doing this, even if abundance estimates are less precise than other methods applied at smaller scales. No-one would doubt that any method that can estimate populations at large scales frequently, accurately and precisely is highly desirable, regardless of the issue of process error. The issue is, are there methods currently available, or can new methods be developed, that meet all the specific requirements of the power predictions (scale, frequency, accuracy, precision). Satellite methods offer this potential, but I don't think the field is at the stage where it is feasible to detect change within just the few years alluded to in lines 102-103 and these threads of text could be taken to imply. It is ambiguous whether this is a theoretical prediction or a prediction that has in mind currently available satellite methods, and this should be clarified. The evidence I am aware of indicates that satellite methods are still too imprecise to be able to detect change within a few years, even at aggregated sites. For example, most of the abundance estimates for multiple aggregated sites derived from high resolution satellite imagery estimates in Lynch and LaRue 2014 have precisions of $\pm 60\%$ which is far too large to detect anything other than a massive change within a few years with high confidence.*

REPLY: We have revised the text in the manuscript to make it clear that presently there are no magic bullets for monitoring these populations regularly at the continental scale (Lines 132–143). We recognize that our power to detect change is limited when using satellite imagery as long as our observation error remains high. Unfortunately, we do not currently have a good handle on the true observation error of high resolution imagery, and the current classification of this imagery as yielding N5 counts is probably too conservative. (In other words, true errors are likely smaller.) Regardless, our model highlights the importance of even crude population estimates (see Lines 139–143) and

for this reason, we think it is important to use all available sources of information (as long as their uncertainties are adequately reflected in the modeling process as is the case with our model). As the technology and our interpretation of satellite imagery matures, we expect that satellite based estimates will exert greater leverage on time series due to their increased precision. In this sense, the inclusion of satellite imagery has future proofed our model.

Reviewer 4

1. *Overall comments I was asked to check the modeling and the coding, both of which I found OK with nothing amiss. In general, the manuscript is well written and the figures illustrate relevant results that are interesting and worthy of publication. The Discussion, though, could be improved in the following ways: The Discussion would read better if its opening paragraph began with the text that is currently on L 120. The opening paragraph could begin with: Our results provide the best understanding of how interannual stochastic variation in seabird population growth rates interact with/influence(?) long-term trends in abundance. If the populations of Adélie penguins are as variable ?..AND CONTINUE TILL THE END OF THE PARAGRAPH*

REPLY: We have re-arranged the discussion as suggested, and moved the text that was originally at the beginning of the discussion further down.

2. *L 109-120 could either be deleted, or follow the above as the second paragraph*

REPLY: See above.

3. *Start a new paragraph on L 192: the new paragraph will be the extinctions para*

REPLY: We have done this; see also response to Query #1.

4. *The Discussion is long. One section that could go is L 209-217. You can start this cropped paragraph at L 218 with something like: The level of process error in a population has important consequences ?TILL END OF THE PARAGRAPH?. You will end up with a short paragraph, but that's OK.*

REPLY: This text addresses some concerns within the penguin community, since the optimal monitoring strategy for CCAMLR is continually under review, and the relative merits of abundance survey data vs. banding for demographic rates is of great interest.

5. *Running the models via someone like Andy Royle (USGS, Patuxent) would be a good idea. Andy is a busy person, so he may agree if its only the models to review.-Another section that could go is L 224-241. It is interesting, but not necessary. Moreover, I am not sure if the statement on L 233-34 “most models for Adélie penguin abundance have not included process error” is correct. Don’t older (non-Bayesian) population models include process error, but not observation error? Don’t hierarchical (Bayesian) models improve on older system process models by allowing the observation process to be modeled separately from the system process, which then allows the introduction of an observation error?*

REPLY: Previous efforts to model Adelie penguin population dynamics have focused primarily on overall population trends (time constant population growth rates) and thus only implicitly include process error as the residual variation of the regression (in this case error does not propagate multiplicatively as does true process error). We have clarified our sentence, which now reads “...have not explicitly included process error...”. We note that while observation error has been noted throughout the history of penguin population census work, very few efforts have been made to either separate process error and observation error, compare their relative magnitude, or consider the implications of large process error for feedback management. Explicitly including process error in a dynamical model, as we have done here, has important implications for future predictions, and more accurately captures the growing uncertainty of predictions in time. (Lynch et al. 2012, which is the precursor to this much more comprehensive analysis, did include both process error and observation error but is restricted in spatial extent and does not address the implications of process error in this way.)

6. *L 252-261 is unnecessary, i.e., delete from: “Noisy” time series TILL END OF PARAGRAPH. IF L 252-261 ARE OMITTED, THEN 1-2 FRESH SENTENCES ARE NEEDED TO END THE DISCUSSION.*

REPLY: We have kept these sentences because they highlight for the general reader aspects of this analysis that apply to other time series. There is a growing body of literature looking at this issue and we think it is important to connect our analysis to this more general discussion in ecology.

7. *Minor edit: In Supplementary 2, given that three chains is mentioned a couple of times in the text, it would be best to change n.chains = 1 in the jags.model call to n.chains = 3.*

REPLY: Because the model is run in parallel, each core has its own chain. If we were to make the edit suggested, we would actually end up with 9 chains in total.

Fig. R1: The finite-population standard deviation for each source of variation in the Adélie intrinsic growth rate that includes random effects for site. For a and b, thick lines represent the 50% equal-tailed credible intervals, thin lines represent the 95% equal-tailed credible intervals, and circles are the posterior medians.

Fig. R2: The finite-population standard deviation for each source of variation in the Adélie intrinsic growth rate in a model that includes random effects for CCAMLR sub-area. For a and b, thick lines represent the 50% equal-tailed credible intervals, thin lines represent the 95% equal-tailed credible intervals, and circles are the posterior medians.

Reviewers' comments:

Reviewer #1 (Remarks to the Author):

I think this manuscript is much improved and I think it is nearly ready for acceptance. My one remaining issue is with the reported trajectories for the South Orkney Islands, particularly given Figure 2a.

For 58.4.1 & 2, 88.1 & 2 & 3, there is clear congruence; there is an apparent hump during the early part of all series, but certainly since 1990 trends have been variable, but increasing.

For 48.1 the trend is more complex; certainly the hump between 1980 and 1990 is evident, though after that the trend is more variable, but ends with an increase.

However, for 48.2, after the 1980-1990 hump, the declining trend is clear and obvious. I am surprised that this major contrast is not picked up in the main text of the paper. It certainly needs comment.

Reviewer #2 (Remarks to the Author):

In my view the points raised in the previous round of review have been satisfactorily addressed in this revision.

Reviewer #3 (Remarks to the Author):

The authors have provided additional information and made revisions that have addressed some of the comments by the reviewers, and in doing so the manuscript has improved. Like the other reviewers, I was concerned about the global model having no spatial term, and while the authors have investigated this further and were unable to improve the model by including such a term, I remain cautious of the utility of the model in its current broad structure. However, the authors have discussed this limitation in the manuscript and on that basis I am willing to accept their treatment of this issue.

One of my main concerns in the previous version of the manuscript related to a regional bias in the data used to parameterise the model. That concern remains despite the inclusion of additional data for the data-sparse East Antarctic region in the revised manuscript. These additional data comprise raw counts of adults from several sites in the Lutzow-Holm Bay region which the authors argue in their response are suitable to estimate population change without bias. The authors give an example in support of their position where colleagues have observed the number of adults being only 5-20% larger than a count of occupied nests, and state that while there is no guarantee the adult-to-nest bias in the inherent data is similarly small, they suggest an 'appropriately timed' adult count will be close to a nest count. However, the timing of the additional counts that were used is largely unknown (almost all the date values for these counts in MAPPPD are NAs) so it is not possible to determine appropriateness, and in the few years where there are dates, they span a period of up to two weeks across November. In this region and across this time period the number of adults present can vary by a factor of up to 100% when the number of nests is relatively constant. This magnitude and variation in bias has the potential to significantly confound estimates of population change. Model predictions are dependent on the accuracy of input data, and in my view these additional data are not appropriate to estimate population change for input to the model unless they are standardised appropriately.

Reviewer 1

1. *I think this manuscript is much improved and I think it is nearly ready for acceptance. My one remaining issue is with the reported trajectories for the South Orkney Islands, particularly given Figure 2a. For 58.4.1-2, 88.1-3, there is clear congruence; there is an apparent hump during the early part of all series, but certainly since 1990 trends have been variable, but increasing. For 48.1 the trend is more complex; certainly the hump between 1980 and 1990 is evident, though after that the trend is more variable, but ends with an increase. However, for 48.2, after the 1980-1990 hump, the declining trend is clear and obvious. I am surprised that this major contrast is not picked up in the main text of the paper. It certainly needs comment.*

REPLY: We added the following sentence to the Results at Line 92 “We also find a long-term decline in abundance in the South Orkney Islands, following an initial period of increase in the early 1980s (Fig. 2).” We added the following sentence to the Discussion at Line 180 “The decline in the South Orkney Islands between 1987 and 2016 is consistent with previous work linking declines at individual sites in this region to bottom up factors resulting from reduced sea ice extent²³.” We have also changed reference 23 from Forcada and Trathan (2009) to Forcada et al. (2006), as this was the original paper that identified population trends in the South Orkney Islands.

Reviewer 3

1. *One of my main concerns in the previous version of the manuscript related to a regional bias in the data used to parameterise the model. That concern remains despite the inclusion of additional data for the data-sparse East Antarctic region in the revised manuscript. These additional data comprise raw counts of adults from several sites in the Lutzow-Holm Bay region which the authors argue in their response are suitable to estimate population change without bias. The authors give an example in support of their position where colleagues have observed the number of adults being only 5-20% larger than a count of occupied nests, and state that while there is no guarantee the adult-to-nest bias in the inherent data is similarly small, they suggest an “appropriately timed” adult count will be close to a nest count. However, the timing of the additional counts that were used is largely unknown (almost all the date values for these counts in MAPPPD are NAs) so it is not possible to determine appropriateness, and in the few years where there are dates, they span a period of up to two weeks across November. In this region and across this time period the number of adults present can vary by a factor of up to 100% when the number of nests is relatively constant. This magnitude and variation in bias has the potential to significantly confound estimates of population change. Model predictions are dependent on the accuracy of input data, and in my view these additional data are not appropriate to estimate population change for input to the model unless they are standardised appropriately.*

REPLY: We disagree with this criticism as our model, by design, separates the measurement error referred to by Reviewer #3 from process error associated with true nest

abundance. In addition, we had purposefully reduced the precision of all adult counts in MAPPPD, such that N1 and N2 reported precisions for adult counts from Kato et al. (2006) for sites in the Lutzow-Holm Bay region (which were included at the request of Reviewer #3 during the previous revision) become N4 and N5 measurement errors, respectively (see Section 1 in Supplementary Methods 1). We should have noted this more explicitly in our initial response letter, and highlight Section IX in Supplementary Methods 1 where our treatment of adult counts is described. Consequently, these adult counts rightfully provide a more uncertain picture of true nest abundances at sites in this area than would have otherwise occurred with comparable time series of nest counts. Nonetheless, the trends observed at these sites are unambiguous (see site-level time series for BENT, HOKU, MAME, MIZU, NOKK, ONGU, RUMP, TOKI, and YTRE in Supplementary Data 10), and given that the time series from Kato et al. (2006) are all relatively complete, we can't imagine a scenario where these trends would emerge purely as an artifact of variable survey timing. We note that our findings are not only consistent with but recapitulate the interpretation of these data as presented in Kato et al. (2006).

REVIEWERS' COMMENTS:

Reviewer #3 (Remarks to the Author):

The authors have presented their case to maintain the use of un-standardized adult counts made largely on unknown dates for the East Antarctic sites added following my first review. I remain of the view that the robustness of the model predictions could be affected by lack of standardization, and that the concepts of bias and precision are different aspects of measurement error that need to be addressed in different ways. The question I am asking myself is how important is this for the overall thesis and conclusions of the paper. I still encourage the authors to address this issue as I think it would remove any doubts. However, in the interest in moving forward on an issue that is important I am willing to accept the authors response in the context of the broad thesis.

Reviewer 3

1. *The authors have presented their case to maintain the use of un-standardized adult counts made largely on unknown dates for the East Antarctic sites added following my first review. I remain of the view that the robustness of the model predictions could be affected by lack of standardization, and that the concepts of bias and precision are different aspects of measurement error that need to be addressed in different ways. The question I am asking myself is how important is this for the overall thesis and conclusions of the paper. I still encourage the authors to address this issue as I think it would remove any doubts. However, in the interest in moving forward on an issue that is important I am willing to accept the authors response in the context of the broad thesis.*

REPLY: We have added the following text to item 2 in Section IX, Additional Assumptions, of Supplementary Data 1 to address the concerns of Reviewer 3 regarding the use of adult counts in the model:

“We recognize that for any individual survey, this may not be the correct conversion factor. Adult counts may be reported when surveys are done very early in the season (prior to nesting), when nests have not yet been established and the count of adults is too small. Alternatively, surveys may be conducted at a time when more than one adult is in attendance at some nests and the count of adults is too high. Because this conversion introduces an additional uncertainty in the conversion to nests, we increased the error category of adult counts by + 3 (with a max error of 5) relative to the reported uncertainty in the adult count itself. This approach is quite conservative, in that we likely overestimate the true uncertainty of those adult counts. Additionally, it is worth noting that the adult counts in our updated database come overwhelmingly (86%) from time series (published by¹⁰ and others) comprised only of adult counts, so any bias (separate from the uncertainty) in this conversion factor (between adults and nests) will not affect our inference regarding the dynamics at these sites, nor the relationship between population growth rate and environmental drivers. Fortunately, the trends observed at these sites are unambiguous (see site-level time series for BENT, HOKU, MAME, MIZU, NOKK, ONGU, RUMP, TOKI, and YTRE in Supplementary Data 10), and given that the time series from¹⁰ are all relatively complete, we can't imagine a scenario where these trends would emerge purely as an artifact of variable survey timing. We note that our findings are not only consistent with, but recapitulate the interpretation of these data as presented in¹⁰. NB: Several of the authors of this paper (Lynch, Jenouvrier, Youngflesh, McDowall) are involved in another project in which UAVs were used to survey both adults and nest counts at the peak of nesting, and we find that a count of adults are only 5-20% larger than a count of occupied nests. While this is no guarantee that the adult-to-nest bias inherent to those included in our model are similarly small, it does suggest

that an appropriately timed adult count is closely related to the number of nesting pairs and so the impact on our estimate of abundance (as an absolute measure) in that region will be minimal.”

Here, reference 10 refers to Kato, A. & Ropert-Coudert, Y. Rapid increase in Adélie penguin populations in Lützow-Holm Bay area since the mid 1990s. *Polar Bioscience* **20**, 55–62 (2006).